# Entropy-Controlled Time-Discretization Bounds and Loss-Adaptive Schedules for Diffusion Models

## Abstract

Diffusion generative models synthesize samples by discretizing reverse-time dynamics driven by a learned score or denoiser. Existing analyses of reverse-diffusion discretization often exhibit explicit dependence on the ambient dimension, while guarantees without explicit dimension dependence typically require structural or geometric assumptions on the target distribution. We develop an information-theoretic approach to reverse-diffusion discretization that avoids such assumptions. We decompose the pathwise KL error into initialization, denoiser approximation, and time-discretization terms, and express the discretization term exactly through the MMSE curve of the associated Gaussian channel. Under finite second moment, countable support, and finite Rényi entropy of order $1/2$, we obtain an entropy-controlled time-discretization bound with no explicit dependence on the ambient dimension. Motivated by the same decomposition, we propose a Loss-Adaptive Schedule (LAS), a practical scheduling rule that uses training-loss information to allocate sampling steps at low post-training cost. Experiments show that LAS is competitive overall and particularly effective in low-step regimes.

## 1 Introduction

Diffusion-based generative models have emerged as one of the most powerful classes of deep generative models (Sohl-Dickstein et al., 2015; Song & Ermon, 2019; Song et al., 2021b), achieving state-of-the-art performance in image (Dhariwal & Nichol, 2021; Ramesh et al., 2022; Rombach et al., 2022), audio (Kong et al., 2021; Liu et al., 2023) and video (Ho et al., 2022) synthesis, as well as molecule generation and protein design (Hoogeboom et al., 2022; Corso et al., 2022; Watson et al., 2023). These methods construct a forward noising process that gradually corrupt data to a prior distribution (usually Gaussian), together with a reverse-time denoising process — either a reverse stochastic differential equation (SDE) (Ho et al., 2020) or a deterministic probability-flow ordinary differential equation (ODE) (Song et al., 2021a) — that transports the prior distribution back to the data distribution. In practice, training a diffusion model amounts to learning the score function of the forward noising process at each noise level; to generate samples, the learned score is used to simulate the reverse-time diffusion process, which is implemented numerically as a finite sequence of denoising steps. Therefore, the main sources of sampling error of a diffusion-based model can be decomposed into statistical error in learning the score function and numerical error from time discretization of the reverse dynamics.

Recently, there has been substantial progress in theoretical understanding of time discretization error of diffusion-model samplers; yet most available guarantees for discretization error still exhibit at least a linear dependence on the ambient dimension $d$ (Benton et al., 2024; Li & Yan, 2025). In contrast, empirically, diffusion models produce high-quality samples with tens to hundreds of steps in very high ambient dimensions. This gap between theory and practice suggests that linear-in-$d$ bounds can be overly conservative.

There are results that improve upon linear-in-$d$ scaling, but they typically come at a cost: they either impose restrictive structural or geometric assumptions on the target distribution, or they establish convergence only in a weaker functional metric than the commonly used total variation (TV) or KL-divergence. For instance, Bruno et al. (2025) derives an error bound of order $\sqrt{d}$ under a log-concavity assumption, while Li et al.

(2025) proves a dimension-free bound assuming the target is well-approximated by a Gaussian mixture. In settings where the data distribution concentrates on a low-dimensional subspace or manifold, Li & Yan (2025); Liang et al. (2025a); Potaptchik et al. (2025) obtain discretization error bounds that scale linearly with an intrinsic dimension rather than the ambient dimension. Finally, de Bortoli et al. (2025) establishes a dimension-free discretization bound, but in a weaker functional metric based on smooth test functionals.

Our main contributions can be summarized as follows:

- We decompose the sampling error in KL divergence into an initialization error term, a score estimation error term and a time discretization error term, and express the discretization error exactly as a minimum mean-square error (MMSE) functional.

- Under finite second moment, countable support, and finite order-1/2 Rényi entropy, we establish an entropy-controlled bound on the time-discretization error with no explicit dependence on the ambient dimension $d$. The learned-denoiser and initialization errors remain separate terms in the KL decomposition.

In addition to these theoretical results, we develop a practical scheduling method motivated by the same KL decomposition:

- We propose a Loss-Adaptive Schedule (LAS) for discretizing the reverse SDE. LAS is a practical scheduling rule that does not require the target distribution to be discrete: it uses an empirical loss profile available from training, or cheaply estimable afterward, to compute a model-adaptive schedule at low post-training cost. Empirically, LAS is competitive overall and gives its strongest improvements in low-step regimes.

## 2    Related Work

Theoretical analysis of diffusion-based generative models typically decomposes sampling error into score estimation error and discretization error from numerically integrating the reverse SDE or its associated probability-flow ODE. For DDPM-style stochastic samplers (Ho et al., 2020) in particular, Chen et al. (2023b) proved one of the first general non-asymptotic guarantees for diffusion models that scales polynomially in problem parameters, without strong assumptions on the data distribution, such as log-concavity and functional inequality. They showed that with $L$-Lipschitzness of score function and a score estimator with $L_2$-error at most $\tilde{O}(\varepsilon)$, a discrete-time reverse diffusion sampler outputs a measure which is $\varepsilon$-close in TV distance to the true data distribution in $\tilde{O}(L^2 d/\varepsilon^2)$ iterations. Chen et al. (2023a) refined this analysis by providing a KL-divergence bound of order $\tilde{O}(d \log(1/\delta)/\varepsilon)$ for the variance-$\delta$ Gaussian perturbation of any data distribution and under a relaxed $1/\delta$-smoothness assumption on score functions.

Later, a number of works sharpen the dimension dependence in these global complexity bounds. Benton et al. (2024) derived the first nearly $d$-linear convergence guarantees via stochastic localization under only finite second-moment assumptions. Li & Yan (2025) proved TV complexity bound of order $\tilde{O}(d/\varepsilon)$ for any target distribution with finite first-order moment. Beyond ambient-dimension-dependent theory, Li & Yan (2025); Liang et al. (2025a); Potaptchik et al. (2025) studied adaptivity to low intrinsic dimension and proved convergence rate of $\tilde{O}(k/\varepsilon)$ in terms of TV and KL-divergence, where $k$ is the intrinsic dimension of the target data distribution.

There are a few recent works regarding dimension independent bounds. Li et al. (2025) obtained TV bound of order $\tilde{O}(1/\varepsilon)$ for data distribution well-approximated by Gaussian mixture models. de Bortoli et al. (2025) derived a dimension-free bound in a weaker functional metric, defined by smooth test functionals with bounded first and second derivatives. Gatmiry et al. (2026) proposed a new collocation-based sampler and proved an iteration complexity logarithmic in $1/\varepsilon$ and no explicit dependence on the ambient dimension; however the dimension enters indirectly through the effective radius of the support of the target distribution.

To date, theoretical bounds on discretization error for diffusion samplers typically scale with the ambient or intrinsic dimension. Existing dimension-free guarantees usually require structural assumptions on the

target distribution, such as mixture structure or other geometric regularity, or else hold in weaker functional metrics. In contrast, we obtain a dimension-free discretization bound through an information-theoretic analysis of the Gaussian channel associated with the forward process. The main quantity governing the bound is the order-1/2 Rényi entropy $H_{1/2}$ of the target distribution. Apart from the basic finite-moment condition needed for the Gaussian channel and KL decomposition, the dimension-free control requires only the finiteness of $H_{1/2}$ and imposes no log-concavity, smoothness, manifold, subspace, or intrinsic-dimension assumptions.

## 3 Problem Setup

Given training samples from a target data distribution $p$, a diffusion model seeks to generate new samples from $p$. It consists of a forward noising process, which progressively corrupts data samples by adding Gaussian noise, and a learned reverse-time denoising process, which synthesizes new data by reversing this corruption and transforming noisy samples back toward the data distribution.

**Forward process** For simplicity, we consider the standard forward diffusion given by Brownian motion started from a data distribution $p$ on $\mathbb{R}^d$:

$$dX_t = dW_t, \qquad X_0 \sim p, \quad t \in [0, T].$$

where $(W_t)_{t \in [0,T]}$ is a Brownian motion on $\mathbb{R}^d$, independent of $X_0$. Throughout, we write $Z := X_0$ for the underlying data sample.

**Reverse process** Let $p_t$ denote the law of $X_t$, and define the (Bayes-optimal) denoiser

$$m_t(x) := \mathbb{E}[Z \mid X_t = x], \qquad t > 0.$$

By Tweedie's formula for Gaussian convolution,

$$\nabla \log p_t(x) = \frac{m_t(x) - x}{t}.$$

Define the reverse-time process $(Y_s)_{s \in [0,T]}$ by $Y_s := X_{T-s}$ for all $s \in [0, T]$. Then, under mild conditions satisfied by the processes considered in Haussmann & Pardoux (1986), the reverse process admits an SDE description

$$dY_s = \beta_s(Y_s) \, ds + dB_s, \qquad Y_0 = X_T,$$

on $s \in [0, T)$, where

$$\beta_s(y) = \nabla \log p_{T-s}(y) = \frac{m_{T-s}(y) - y}{T - s},$$

and $(B_s)_{s \in [0,T]}$ is a Brownian motion.

**Approximate reverse process** In practice, this reverse-time process is implemented on a discretized time grid, and the unknown denoiser $m$ is replaced by an approximation $\hat{m}$ obtained from a learned score function.

Fix a small $\delta > 0$ and let $T_\delta := T - \delta$. Fix a time grid

$$0 = s_0 < s_1 < \cdots < s_K = T_\delta < T.$$

We now define an approximate reverse process $(\tilde{Y}_s)_{s \in [0,T_\delta]}$ by replacing the true denoiser $m_{T-s}$ with an approximation $\hat{m}_{T-s_{k-1}}$ whose time index is frozen on each interval, and whose value is evaluated at the previous gridpoint state. In particular, we keep all other terms in the drift of reverse process unchanged. Concretely, for $s \in (s_{k-1}, s_k]$ we use the drift

$$\tilde{\beta}_s^{(k)}(y) := \frac{\hat{m}_{T-s_{k-1}}(Y_{s_{k-1}}) - y}{T - s}.$$

On the discretization grid, we have

$$\tilde{Y}_{s_k} - \tilde{Y}_{s_{k-1}} = \int_{s_{k-1}}^{s_k} \tilde{\beta}_s^{(k)}(\tilde{Y}_s)\, ds + (B_{s_k} - B_{s_{k-1}}).$$

for $k = 1, \ldots, K$.

The approximate reverse process $(\tilde{Y}_s)_{s \in [0, T_\delta]}$ satisfies

$$d\tilde{Y}_s = \tilde{\beta}_s(\tilde{Y}_s)\, ds + dB_s, \qquad \tilde{Y}_0 \sim q,$$

on $s \in [0, T_\delta]$, where $\tilde{\beta}_s(y) = \tilde{\beta}_s^{(k)}(y)$, for $s \in (s_{k-1}, s_k]$ and $k = 1, \ldots, K$. The processes $(Y, \tilde{Y})$ are coupled using the same Brownian increments on each interval. Since the distribution of $X_T$ is not available, in practice, one initializes the approximate reverse process with a tractable prior $q$. For the Brownian forward process, one typically chooses $q$ to be $\mathcal{N}(0, T I_d)$.

**Error decomposition**    Finally, we express the sampling error in terms of the KL divergence between the sampling distribution and the target distribution.

On each interval $(s_{k-1}, s_k]$ define the drift mismatch

$$\delta_s := \beta_s(Y_s) - \tilde{\beta}_s^{(k)}(Y_s).$$

Let $\mathbb{P}$ and $\tilde{\mathbb{P}}$ denote the path laws of $Y$ and $\tilde{Y}$ on $C([0, T_\delta], \mathbb{R}^d)$.

**Proposition 3.1.** *Assume the square-integrability condition*

$$\mathbb{E}_{\mathbb{P}}\left[\int_0^{T_\delta} \|\delta_s\|^2 ds\right] < \infty. \tag{1}$$

*The total pathwise KL has upper bound*

$$\begin{aligned}
\mathrm{KL}(\mathbb{P}\|\tilde{\mathbb{P}}) &\leq \mathrm{KL}(p_T\|q) + \frac{1}{2}\, \mathbb{E}_{\mathbb{P}}\left[\int_0^{T_\delta} \|\delta_s\|^2\, ds\right] \\
&= \mathrm{KL}(p_T\|q) + \frac{1}{2}\sum_{k=1}^{K} \mathbb{E}_{\mathbb{P}}\left[\int_{s_{k-1}}^{s_k} \left\|\beta_s(Y_s) - \tilde{\beta}_s^{(k)}(Y_s)\right\|^2 ds\right].
\end{aligned} \tag{2}$$

*Remark* 3.2. (1) is the natural condition for the right-hand side of the desired bound to be finite, and ensures that the stochastic integral $\int_0^t \delta_s^\top dB_s$ is well-defined on $[0, T_\delta]$.

Equation (2) controls the discrepancy between the path laws of the exact and approximate reverse processes. Our ultimate goal, however, is to control the discrepancy between the terminal sampling distributions at time $T_\delta$ (i.e., the laws of $Y_{T_\delta}$ and $\tilde{Y}_{T_\delta}$). Since this map is measurable, the data processing inequality for KL divergence gives

$$\mathrm{KL}(\mathbb{P}_{T_\delta} \,\|\, \tilde{\mathbb{P}}_{T_\delta}) \leq \mathrm{KL}(\mathbb{P} \,\|\, \tilde{\mathbb{P}}).$$

Therefore, any upper bound on the pathwise KL in (2) immediately yields an upper bound on the KL divergence between the sampling distribution produced by the discretized reverse dynamics and the true reverse-time marginal at time $T_\delta$. In particular, it suffices to bound the right-hand side of (2).

We now split the drift mismatch into two contributions: (i) the error from freezing the time index and using the previous gridpoint state, and (ii) the error from replacing the true denoiser by $\hat{m}$.

Insert and subtract $m_{T-s_{k-1}}(Y_{s_{k-1}})$:

$$\beta_s(Y_s) - \tilde{\beta}_s^{(k)}(Y_s) = \frac{m_{T-s}(Y_s) - m_{T-s_{k-1}}(Y_{s_{k-1}})}{T-s} + \frac{m_{T-s_{k-1}}(Y_{s_{k-1}}) - \hat{m}_{T-s_{k-1}}(Y_{s_{k-1}})}{T-s}.$$

Denote $t := T - s$ and $t_{k-1} := T - s_{k-1}$. Let $\mathcal{F}_t$ be the filtration defined by $\mathcal{F}_t = \sigma(X_u : t \leq u \leq T)$. For $t \leq t_{k-1}$, since $\mathcal{F}_{t_{k-1}} \subset \mathcal{F}_t$, using the tower property and Markov property, we have

$$\mathbb{E}\big[m_t(X_t) - m_{t_{k-1}}(X_{t_{k-1}}) \mid \mathcal{F}_{t_{k-1}}\big] = 0.$$

Since $(Y_s)_{s \in [0, T_\delta]} = (X_{T-s})_{s \in [0, T_\delta]}$, the cross term vanishes when decomposing the quadratic drift mismatch. Thus the quadratic term in (2) splits exactly into discretization and approximation contributions:

$$\sum_{k=1}^{K} \mathbb{E}_{\mathbb{P}}\left[\int_{s_{k-1}}^{s_k} \left\|\beta_s(Y_s) - \tilde{\beta}_s^{(k)}(Y_s)\right\|^2 ds\right] = \mathcal{E}_{\text{disc}} + \mathcal{E}_{\text{apx}}, \tag{3}$$

and the KL is consequently upper bounded by

$$\text{KL}(\mathbb{P}_{T_\delta} \| \tilde{\mathbb{P}}_{T_\delta}) \leq \mathcal{E}_{\text{init}} + \frac{1}{2}(\mathcal{E}_{\text{disc}} + \mathcal{E}_{\text{apx}}), \tag{4}$$

where

$$\begin{aligned}
\mathcal{E}_{\text{init}} &:= \text{KL}(p_T \| q), \\
\mathcal{E}_{\text{disc}} &:= \sum_{k=1}^{K} \mathbb{E}_{\mathbb{P}}\left[\int_{s_{k-1}}^{s_k} \frac{\left\|m_{T-s}(Y_s) - m_{T-s_{k-1}}(Y_{s_{k-1}})\right\|^2}{(T-s)^2} ds\right], \\
\mathcal{E}_{\text{apx}} &:= \sum_{k=1}^{K} \mathbb{E}_{\mathbb{P}}\left[\int_{s_{k-1}}^{s_k} \frac{\left\|m_{T-s_{k-1}}(Y_{s_{k-1}}) - \hat{m}_{T-s_{k-1}}(Y_{s_{k-1}})\right\|^2}{(T-s)^2} ds\right].
\end{aligned} \tag{5}$$

This controls the terminal sampling error relative to the Gaussian-smoothed distribution $\mathbb{P}_{T_\delta} = p_\delta = p * \mathcal{N}(0, \delta I_d)$. It does not directly control the KL divergence relative to the original distribution $p$. The discrepancy between $p_\delta$ and $p$ is a separate endpoint-smoothing issue and is beyond the scope of the present work.

The term $\mathcal{E}_{\text{init}}$ comes from initializing the reverse process with $q$ instead of the exact $p_T$. The term $\mathcal{E}_{\text{apx}}$ is driven entirely by the quality of the estimator $\hat{m}$ (equivalently, the learned score), and corresponds to the statistical error. The term $\mathcal{E}_{\text{disc}}$ is the numerical time discretization error of the reverse dynamics; it persists even if one had access to the exact denoiser, and it is the object of our main entropy-controlled discretization analysis.

## 4 Main Results

In this section, we state our main results. Our first goal is to rewrite the discretization error $\mathcal{E}_{\text{disc}}$ in a functional form that depends only on the MMSE along the forward Gaussian channel. This representation turns the discretization analysis into a problem of controlling how the MMSE varies with the SNR. Our second goal is to obtain an entropy-controlled upper bound on $\mathcal{E}_{\text{disc}}$ with no explicit dependence on the ambient dimension by proving an explicit bound on the derivative of the MMSE. Throughout, we work under mild conditions on the target distribution, stated next.

**Assumption 4.1.** For $Z \sim p$, we have $\mathbb{E}\|Z\|^2 = M_2 < \infty$.

**Assumption 4.2.** Suppose the target data distribution $p$ is discrete and supported on a countable set $\mathcal{C} \subset \mathbb{R}^d$.

*Remark* 4.3. Assumption 4.2 models the target distribution $p$ as discrete on a countable subset of $\mathbb{R}^d$. This matches a common and practically relevant setting in diffusion modeling: latent diffusion models (LDMs) built on a vector-quantized (VQ) first stage. In VQ-based representations, each latent is obtained by selecting codebook indices from a finite set and mapping them to real-valued codebook embeddings. Although the embeddings are vectors in $\mathbb{R}^d$, the latent variable itself takes values in a finite (hence countable) subset of $\mathbb{R}^d$.

*Remark* 4.4. The class-conditional ImageNet LDM-VQ-8 model used in our experiments provides a concrete example of Assumption 4.2. Its first stage is VQ-regularized with downsampling factor $f = 8$, so a $256 \times 256$ image is represented by a $32 \times 32$ grid of

$$L = 32 \cdot 32 = 1024$$

discrete code indices. The codebook contains

$$S = 16384 = 2^{14}$$

entries, each mapped to an embedding in $\mathbb{R}^4$. Thus, if

$$J = (J_1, \ldots, J_{1024}) \in [16384]^{1024}$$

denotes the complete code sequence and $Z = \Phi(J)$ its flattened embedding, then

$$Z \in \mathbb{R}^{32 \cdot 32 \cdot 4} = \mathbb{R}^{4096},$$

but $Z$ still has finite support. In particular,

$$|\operatorname{supp}(Z)| \leq S^L = 16384^{1024} = 2^{14336}.$$

Consequently, in bits,

$$H_{1/2}^{(2)}(Z) \leq 14336,$$

or equivalently $H_{1/2}(Z) \leq 14336 \log 2$ under the natural-log convention of Definition 4.7. This calculation verifies the theorem-level connection to the experimental latent model: although the embedded variable lies in $\mathbb{R}^{4096}$, the entropy bound is determined by the underlying discrete law rather than directly by the number of embedding coordinates. The value 14336 is only a worst-case support bound, however, because it treats every code sequence as attainable and ignores dependence and nonuniformity among latent positions.

A representation-based way to assess the entropy scale of ImageNet is to examine a particularly compact, high-fidelity discrete representation. If $C = E(X)$ is an injective encoding on the support of an image-valued random variable $X$, then the encoding only relabels the outcomes and

$$H_{1/2}(C) = H_{1/2}(X).$$

Thus, when a compact tokenizer is approximately information-preserving at a given reconstruction fidelity, the capacity of its code space provides a practical upper proxy for the effective entropy of the represented image distribution at that fidelity.

TiTok-L-32 represents an ImageNet $256 \times 256$ image using only 32 discrete tokens from a codebook of size $4096 = 2^{12}$ (Yu et al., 2024). Its complete code space therefore contains at most

$$4096^{32} = (2^{12})^{32} = 2^{384}$$

elements, and hence

$$H_{1/2}^{(2)}(Z_{\text{TiTok}}) \leq 384 \quad \text{bits.}$$

Moreover, because the TiTok decoder is deterministic, the distribution of decoded images also has support cardinality at most $2^{384}$ and therefore satisfies the same entropy upper bound. Since TiTok achieves high-quality ImageNet reconstruction and generation with this compact representation, we use 384 bits as an empirically grounded upper proxy for the effective entropy of ImageNet at TiTok's reconstruction fidelity. This is substantially sharper than the naive 14336-bit support bound for the experimental VQ grid.

*Remark* 4.5. Assumption 4.2 is used only for the entropy-based MMSE derivative bound and the resulting entropy-controlled discretization theorem. It is not an assumption required by the Loss-Adaptive Schedule introduced in Section 5. LAS is an algorithmic scheduling rule based on an empirical loss profile over noise levels, and can therefore be applied to standard continuous or latent diffusion models whenever such a loss profile is available.

**Discretization error as an MMSE functional**  Recall that the forward process is Brownian motion started from the data: $X_t = Z + W_t$, $t \in [0, T]$, with $Z := X_0 \sim p$, where $(W_t)_{t \in [0,T]}$ is independent of $Z$. Let $m_t(x) := \mathbb{E}[Z \mid X_t = x]$ denote the Bayes-optimal denoiser at noise level $t$. We measure the denoising error through the minimum mean-squared error (MMSE) along the Gaussian channel, parameterized by the signal-to-noise ratio (SNR) $\gamma := 1/t$:

$$\mathrm{mmse}(\gamma) := \mathbb{E}\big[\|Z - m_{1/\gamma}(X_{1/\gamma})\|_2^2\big], \qquad \gamma > 0.$$

Let $\{s_k\}_{k=0}^K$ be the reverse-time grid on $[0, T - \delta]$ and define the corresponding SNR grid

$$\gamma_k := \frac{1}{T - s_k}, \qquad k = 0, 1, \ldots, K,$$

so that $\gamma_0 = 1/T$ and $\gamma_K = 1/\delta$. Our first step is to express the reverse-time discretization error $\mathcal{E}_{\mathrm{disc}}$ (defined in (5)) as a functional of $\mathrm{mmse}(\cdot)$.

**Proposition 4.6.** *Under assumption 4.1 discretization error $\mathcal{E}_{\mathrm{disc}}$ in (5) satisfies*

$$\mathcal{E}_{\mathrm{disc}} = \sum_{k=1}^K \int_{\gamma_{k-1}}^{\gamma_k} \big(\mathrm{mmse}(\gamma_{k-1}) - \mathrm{mmse}(\gamma)\big)\, d\gamma. \tag{6}$$

Identity (6) shows that $\mathcal{E}_{\mathrm{disc}}$ is the cumulative *area gap* between $\mathrm{mmse}(\gamma)$ and its left-endpoint values over each SNR interval. This representation reduces control of $\mathcal{E}_{\mathrm{disc}}$ to understanding $\mathrm{mmse}(\gamma)$.

**From MMSE to an entropy-controlled bound**  Since $\mathrm{mmse}(\gamma)$ is nonincreasing in $\gamma$, we can bound the area gap on each interval using the slope of mmse:

$$\mathrm{mmse}(\gamma_{k-1}) - \mathrm{mmse}(\gamma) \leq (\gamma - \gamma_{k-1}) \sup_{\xi \in [\gamma_{k-1}, \gamma]} (-\mathrm{mmse}'(\xi)), \tag{7}$$

and integrating (7) over $\gamma \in [\gamma_{k-1}, \gamma_k]$ yields

$$\mathcal{E}_{\mathrm{disc}} \leq \sum_{k=1}^K \frac{(\Delta \gamma_k)^2}{2} \sup_{\gamma \in [\gamma_{k-1}, \gamma_k]} \big(-\mathrm{mmse}'(\gamma)\big), \tag{8}$$

where $\Delta \gamma_k := \gamma_k - \gamma_{k-1}$.

The key technical input is a bound for the MMSE derivative based on the Rényi entropy.

**Definition 4.7** (Rényi entropy of order 1/2)**.** For a discrete distribution $p$ supported on $\mathcal{C}$, define

$$H_{1/2} := \frac{1}{1 - \frac{1}{2}} \log \sum_{z \in \mathcal{C}} p(z)^{1/2} = 2 \log \sum_{z \in \mathcal{C}} \sqrt{p(z)}.$$

**Theorem 4.8.** *Suppose Assumptions 4.1 and 4.2 hold. Then for all $\gamma > 0$,*

$$|\mathrm{mmse}'(\gamma)| \leq \frac{\Psi(H_{1/2})}{\gamma^2}, \tag{9}$$

*where $\Psi(h) := 96h^2 + 64(h + 1)$.*

**Corollary 4.9.** *If $H_{1/2}(p) \geq 2$, then*

$$|\mathrm{mmse}'(\gamma)| \leq \frac{144 H_{1/2}^2}{\gamma^2}.$$

Combining (8) and (9) gives

$$\mathcal{E}_{\mathrm{disc}} \leq \frac{\Psi(H_{1/2})}{2} \sum_{k=1}^K \frac{(\Delta \gamma_k)^2}{\gamma_{k-1}^2}. \tag{10}$$

*Remark* 4.10. The term dimension-free refers to the absence of an explicit dependence on the ambient Euclidean dimension $d$ in the discretization bound. The bound is not distribution-free: its complexity is captured by information-theoretic quantities such as $H_{1/2}$. Thus, the result should be interpreted as replacing ambient-dimensional dependence by entropy dependence.

With further assumptions, we can bound Rényi entropy with Shannon entropy up to a constant, and this gives a bound for $|\text{mmse}'(\gamma)|$ and hence $\mathcal{E}_{\text{disc}}$ in terms of the Shannon entropy as a corollary.

**Definition 4.11.** For a probability density function or probability mass function $p$ supported on $\mathcal{C} \subset \mathbb{R}^d$, define the information content for any $z \in \mathcal{C}$ as

$$\imath(z) := \log \frac{1}{p(z)}.$$

Define the Shannon entropy of $p$ as

$$H := \mathbb{E}_{Z \sim p}[\imath(Z)].$$

**Assumption 4.12.** Assume the information content is sub-exponential about its mean, *i.e.* there exist constants $\nu^2 > 0$ and $b \in (0, 2]$ such that for all $\lambda \in \mathbb{R}$ with $|\lambda| \le 1/b$,

$$\mathbb{E} \exp\left(\lambda(\imath(Z) - H)\right) \le \exp\left(\nu^2 \lambda^2\right). \tag{SE}$$

*Remark* 4.13. Assumption 4.12 is *purely information-theoretic*: it constrains only the fluctuations of the information content $\imath(Z) = -\log p(Z)$ around its mean $H$, where $Z \sim p$. In particular, it imposes *no* geometric or norm-based regularity on $p$—for example, it does not assume log-concavity, smoothness, manifold/subspace structure or intrinsic dimension.

**Corollary 4.14.** *Under Assumptions 4.1, 4.2, and 4.12, for all $\gamma > 0$,*

$$|\text{mmse}'(\gamma)| \le \frac{\Psi_\nu(H)}{\gamma^2}, \qquad \mathcal{E}_{\text{disc}} \le \frac{\Psi_\nu(H)}{2} \sum_{k=1}^{K} \frac{(\Delta\gamma_k)^2}{\gamma_{k-1}^2}, \tag{11}$$

*where*

$$\Psi_\nu(H) := 32\left[3\left(H + \frac{\nu^2}{2}\right)^2 + 2\left(H + \frac{\nu^2}{2}\right) + 2\right].$$

**Choosing the SNR grid: geometric spacing** Given (10), we obtain an upper bound for $\mathcal{E}_{\text{disc}}$ by optimizing over all the SNR grid $\{\gamma_k\}$ subject to fixed endpoints. Let $r_k := \gamma_k/\gamma_{k-1} > 0$. Then $\Delta\gamma_k = \gamma_{k-1}(r_k - 1)$ and hence

$$\frac{(\Delta\gamma_k)^2}{\gamma_{k-1}^2} = (r_k - 1)^2.$$

Moreover the endpoint constraint becomes

$$\prod_{k=1}^{K} r_k = \frac{\gamma_K}{\gamma_0} = \frac{T}{\delta} =: \Lambda.$$

Therefore, minimizing the bound (10) reduces to

$$\min\left\{ \sum_{k=1}^{K}(r_k - 1)^2 : r_k > 0, \prod_{k=1}^{K} r_k = \Lambda \right\}.$$

By symmetry (and convexity of $x \mapsto (e^x - 1)^2$ after the change of variables $x_k = \log r_k$), the minimum is attained when all ratios are equal, $r_k \equiv r$, hence $r^K = \Lambda$ and $r = \Lambda^{1/K}$. Equivalently, the optimal grid is *geometric* or *log-linear* in SNR:

$$\gamma_k = \gamma_0 \Lambda^{k/K}, \qquad k = 0, 1, \dots, K. \tag{12}$$

We remark that this coincides with the widely used "log SNR" discretization heuristic in diffusion sampling, and here it emerges as the minimizer of the upper bound.

For this choice of SNR grid,

$$\sum_{k=1}^{K} \frac{(\Delta \gamma_k)^2}{\gamma_{k-1}^2} = \sum_{k=1}^{K} (\Lambda^{1/K} - 1)^2 = K(\Lambda^{1/K} - 1)^2,$$

and from (10) we conclude the entropy-controlled discretization bound

$$\mathcal{E}_{\mathrm{disc}} \leq \frac{\Psi(H_{1/2})}{2} K(\Lambda^{1/K} - 1)^2. \tag{13}$$

The same geometric grid also yields a clean expression for the statistical (approximation) term $\mathcal{E}_{\mathrm{apx}}$ in (4) when the learned model is parameterized as an $\varepsilon$-predictor. For $\gamma > 0$, write the Gaussian channel as

$$X_{1/\gamma} = Z + \frac{1}{\sqrt{\gamma}}\, \varepsilon, \qquad Z := X_0 \sim p, \;\; \varepsilon \sim \mathcal{N}(0, I_d),$$

and define the Bayes-optimal noise predictor

$$\varepsilon_\gamma^\star(x) := \sqrt{\gamma}\big(x - m_{1/\gamma}(x)\big) = \mathbb{E}\big[\varepsilon \mid X_{1/\gamma} = x\big].$$

Given any learned predictor $\hat{\varepsilon}_\gamma(\cdot)$, define the induced learned denoiser

$$\hat{m}_{1/\gamma}(x) := x - \frac{1}{\sqrt{\gamma}}\, \hat{\varepsilon}_\gamma(x).$$

**Proposition 4.15.** *Let $\{\gamma_k\}_{k=0}^{K}$ be the SNR grid (12). Define the per-level $\varepsilon$-prediction MSE*

$$\epsilon_k := \mathbb{E}\Big[\big\|\varepsilon_{\gamma_{k-1}}^\star(X_{1/\gamma_{k-1}}) - \hat{\varepsilon}_{\gamma_{k-1}}(X_{1/\gamma_{k-1}})\big\|_2^2\Big]$$

*for $k = 1, \ldots, K$. Then*

$$\mathcal{E}_{\mathrm{apx}} = \big(\Lambda^{1/K} - 1\big) \cdot \sum_{k=1}^{K} \epsilon_k. \tag{14}$$

Combining Proposition 4.15 with the discretization bound (13) yields

$$\mathrm{KL}(\mathbb{P}\|\tilde{\mathbb{P}}) = \mathcal{E}_{\mathrm{init}} + \frac{1}{2}\big(\mathcal{E}_{\mathrm{disc}} + \mathcal{E}_{\mathrm{apx}}\big)$$

$$\leq \mathcal{E}_{\mathrm{init}} + \frac{K}{2}\big(\Lambda^{1/K} - 1\big)\left[\frac{\Psi(H_{1/2})}{2}\big(\Lambda^{1/K} - 1\big) + \frac{1}{K}\sum_{k=1}^{K} \epsilon_k\right].$$

In particular, if $K \geq \log \Lambda$, then

$$\Lambda^{1/K} = e^{(\log \Lambda)/K} \leq 1 + 2\frac{\log \Lambda}{K}.$$

Combining this estimate with the pathwise KL bound gives

$$\mathrm{KL}(\mathbb{P}\|\tilde{\mathbb{P}}) \leq \mathcal{E}_{\mathrm{init}} + \log \Lambda \left[\frac{\Psi(H_{1/2})}{K} \log \Lambda + \frac{1}{K}\sum_{k=1}^{K} \epsilon_k\right]. \tag{15}$$

We now account for the initialization mismatch in practical sampling. The exact reverse process is initialized from the law $p_T$ of

$$X_T = Z + \sqrt{T}\epsilon, \qquad \epsilon \sim \mathcal{N}(0, I_d),$$

whereas the implemented sampler starts from the tractable prior $q = \mathcal{N}(0, TI_d)$. Since

$$p_T = \int \mathcal{N}(z, TI_d)\, p(dz),$$

convexity of KL divergence gives

$$\mathrm{KL}(p_T \,\|\, q) \leq \mathbb{E}_Z\, \mathrm{KL}(\mathcal{N}(Z, TI_d) \,\|\, \mathcal{N}(0, TI_d)) = \frac{\mathbb{E}\|Z\|^2}{2T} = \frac{\gamma_0}{2} M_2. \tag{16}$$

Therefore, by the chain rule for pathwise KL and the data-processing inequality,

$$\mathrm{KL}(\mathbb{P}_{T_\delta} \,\|\, \tilde{\mathbb{P}}_{T_\delta}) \leq \frac{\gamma_0}{2} M_2 + \log \Lambda \left[ \frac{\Psi(H_{1/2})}{K} \log \Lambda + \frac{1}{K} \sum_{k=1}^{K} \epsilon_k \right]. \tag{17}$$

*Remark* 4.16 (Comparison with discrete diffusion samplers for language). Consider a diffusion language model in which a length-$L$ token sequence

$$J = (J_1, \ldots, J_L) \in [S]^L$$

over a vocabulary of size $S$ is embedded into Euclidean space by a deterministic map $Z = \Phi(J)$, and the diffusion is then performed by Gaussian noising in the embedding space. Since $Z$ is a deterministic function of $J$,

$$H_{1/2}(Z) \leq H_{1/2}(J) \leq \log |[S]^L| = L \log S.$$

Therefore, ignoring endpoint and approximation factors, for sufficiently large Rényi entropy, our worst-case discretization bound gives

$$\mathcal{E}_{\mathrm{disc}} \lesssim \frac{H_{1/2}(Z)^2}{K} \leq \frac{L^2 (\log S)^2}{K}.$$

Equivalently, to make the discretization error at most $\varepsilon$, the worst-case step complexity is

$$K = \widetilde{O}\left( \frac{L^2 (\log S)^2}{\varepsilon} \right).$$

This should be contrasted with convergence guarantees for discrete diffusion samplers that act directly on the token space. For example, the recent $\tau$-leaping analysis of Liang et al. (2025b) gives a step complexity of order

$$\widetilde{O}\left( \frac{L^2 S}{\varepsilon} \right),$$

where $L$ is the sequence length and $S$ is the vocabulary size. Thus, in the Gaussian-embedding setting considered here, the worst-case dependence on sequence length is of the same quadratic order in $L$, but the dependence on vocabulary size improves from linear in $S$ to quadratic in $\log S$. Moreover, this is only a worst-case entropy bound: for natural language distributions, $H_{1/2}(J)$ may be much smaller than $L \log S$, in which case the entropy-based bound can be substantially sharper.

## 5 Do Not Throw Away the Training Loss!

In (17) we see a limitation of choosing the schedule by analyzing $\mathcal{E}_{\mathrm{disc}}$ alone: even if geometric spacing is optimal for our upper bound on discretization, the *total* error that matters in practice is $\mathcal{E}_{\mathrm{disc}} + \mathcal{E}_{\mathrm{apx}}$, and the approximation term depends on how the model error $\epsilon_k$ is distributed across noise levels. This suggests that the best sampling schedule should not be universal, but instead adapt to the trained model.

The theory and LAS therefore play complementary roles. Section 4 controls the intrinsic discretization term $\mathcal{E}_{\mathrm{disc}}$ and shows that geometric SNR spacing is optimal for the resulting entropy-based upper bound. In contrast, the learned-model setting also contains the approximation term $\mathcal{E}_{\mathrm{apx}}$, whose contribution depends

on the trained model's error profile. LAS is motivated by the same KL decomposition, but it targets the schedule-dependent sum $\mathcal{E}_{\mathrm{disc}} + \mathcal{E}_{\mathrm{apx}}$ rather than following directly from the Rényi-entropy bound.

In this section, we take this perspective seriously and ask: *given a trained diffusion model, can we use the information already present in its training loss to choose a better discretization schedule?*

Importantly, the resulting scheduling rule does not require the target distribution to be discrete. The discreteness assumption in Section 4 is used for the theoretical entropy-based control of the MMSE derivative. In contrast, LAS only uses an empirical estimate of the model's loss profile across noise levels. Therefore, the schedule can be applied directly to continuous-state or latent diffusion models, including the ImageNet latent diffusion experiments below.

We rewrite $\mathcal{E}_{\mathrm{disc}} + \mathcal{E}_{\mathrm{apx}}$ in terms of $x_0$-prediction risks—quantities directly tied to standard $\varepsilon$-training losses— and an MMSE functional. This motivates LAS as a primarily practical, model-adaptive scheduling method derived from our loss decomposition. LAS uses a loss profile already available from training, or cheaply estimated afterward, and computes the schedule through a finite-grid dynamic-programming problem. We view this as a lightweight alternative to Align Your Steps (Sabour et al., 2024), which requires a separate post-training procedure involving Monte Carlo estimation and schedule optimization. Numerical experiments show that LAS is competitive overall, with particularly strong performance in low-step regimes.

The next proposition shows that the sum $\mathcal{E}_{\mathrm{disc}} + \mathcal{E}_{\mathrm{apx}}$ can be rewritten in terms of the model's $x_0$-prediction risk, a quantity that is already available (or can be cheaply estimated) at the end of training.

**Proposition 5.1.** *For each $k = 1, \ldots, K$, define the model $x_0$-prediction risk at SNR $\gamma_{k-1}$ by*

$$\mathcal{L}_{x_0}(\gamma_{k-1}) := \mathbb{E}\left[\left\|Z - \hat{m}_{1/\gamma_{k-1}}(X_{1/\gamma_{k-1}})\right\|_2^2\right].$$

*Then, under Assumption 4.1, $\mathcal{E}_{\mathrm{disc}} + \mathcal{E}_{\mathrm{apx}}$ is equal to*

$$\sum_{k=1}^{K}(\gamma_k - \gamma_{k-1})\,\mathcal{L}_{x_0}(\gamma_{k-1}) \;-\; \int_{\gamma_0}^{\gamma_K} \mathrm{mmse}(\gamma)\,d\gamma. \tag{18}$$

A key feature of (18) is that the integral term depends only on the endpoints and is therefore independent of the discretization schedule. Consequently, for fixed $K$ and fixed endpoints, minimizing $\mathcal{E}_{\mathrm{disc}} + \mathcal{E}_{\mathrm{apx}}$ (and hence tightening the KL control) is equivalent to minimizing the weighted sum

$$\min_{\gamma_0 < \gamma_1 < \cdots < \gamma_K} \; \sum_{k=1}^{K}(\gamma_k - \gamma_{k-1})\,\mathcal{L}_{x_0}(\gamma_{k-1}).$$

Note the quantity controlled by $\mathcal{E}_{\mathrm{disc}} + \mathcal{E}_{\mathrm{apx}}$ is a pathwise KL divergence, and we only access the terminal discrepancy through the data-processing inequality; consequently, directly optimizing the bound can be inefficient. Empirically, the looseness is most pronounced at large SNR (near $\gamma_K = 1/\delta$), where the pathwise KL can overweight fine-scale, near-terminal errors that do not translate proportionally into terminal sample quality. We provide evidence for this phenomenon in Appendix I. In particular, the ablations show that schedules which place too many steps in the near-terminal low-noise region need not yield the best sample quality, even though this region is strongly weighted by the pathwise KL/loss-based objective. This supports our use of a regularized SNR-axis in LAS: rather than following the raw loss profile alone, LAS balances loss adaptivity with a smoother allocation of steps across noise levels.

Accordingly, we optimize the schedule on a regularized SNR axis that compresses the high-SNR regime, where the pathwise KL upper bound can be overly sensitive. For a parameter $\lambda > 0$, define

$$\gamma_{\mathrm{reg}}(\gamma) \; := \; \frac{\gamma}{1 + \lambda^2 \gamma}.$$

The particular form of $\gamma_{\mathrm{reg}}$ is a practical design choice rather than a consequence of Theorem 4.8 or Proposition 5.1: it is monotone, approximately linear at low SNR, smoothly saturates at high SNR, and introduces

Table 1: ImageNet $256 \times 256$ metrics (FID, sFID, IS).

| Sampler | Schedule | NFE=10 | | | NFE=20 | | |
|---|---|---|---|---|---|---|---|
| | | FID ↓ | sFID ↓ | IS ↑ | FID ↓ | sFID ↓ | IS ↑ |
| DDIM ($\eta = 1$) | LAS | **15.71** | **47.79** | **168.94** | **8.56** | **15.33** | **282.89** |
| | Time-uniform | 25.06 | 68.56 | 111.78 | 9.44 | 22.84 | 255.74 |
| | LogSNR | 68.89 | 130.27 | 24.01 | 16.02 | 46.19 | 166.02 |
| | EDM ($\rho = 7$) | 66.72 | 127.00 | 25.03 | 17.42 | 49.86 | 152.38 |
| SDE-DPM++ (2M) | LAS | **6.20** | **10.59** | **273.76** | 6.67 | **6.18** | 320.18 |
| | Time-uniform | 7.94 | 16.39 | 242.81 | 7.65 | 7.00 | **320.83** |
| | LogSNR | 10.54 | 29.60 | 191.28 | 7.15 | 8.56 | 308.36 |
| | EDM ($\rho = 7$) | 9.91 | 27.23 | 199.77 | 7.36 | 8.74 | 296.43 |
| DPM++ (2M) | LAS | **4.59** | **5.74** | **263.69** | 4.84 | **5.37** | 283.09 |
| | Time-uniform | 5.53 | 6.78 | 243.78 | 5.48 | 5.44 | **284.85** |
| | LogSNR | 4.95 | 6.93 | 251.08 | 4.98 | 5.47 | 280.89 |

only one scale parameter. Thus it preserves the ordering of noise levels while downweighting very large SNR values (near the terminal part of the reverse process), which are precisely where the pathwise-KL surrogate is typically loosest. Appendix I reports sensitivity checks for both $\lambda$ and the second-order smoothness parameter $\alpha$.

With $\eta_k := \gamma_{\text{reg}}(\gamma_k)$, we replace the unregularized objective by the surrogate

$$\min_{\gamma_0 < \gamma_1 < \cdots < \gamma_K} \sum_{k=1}^{K} \left( \eta_k - \eta_{k-1} \right) \mathcal{L}_{x_0}(\gamma_{k-1}) \tag{19}$$

That is, we keep evaluating the model diagnostic at the *true* SNR points $\gamma_{k-1}$, but we measure step sizes on the *regularized* axis via $\Delta \eta_k = \eta_k - \eta_{k-1}$, preventing near-terminal (high-SNR) steps from dominating the optimization.

Once $\mathcal{L}_{x_0}(\gamma)$ is estimated on a finite candidate set of SNR values, the minimization in (19) becomes a minimum-cost selection of $K$ increasing grid points and can be solved efficiently by a standard dynamic-programming shortest-path routine (details provided in Appendix G). We call this schedule Loss-Adaptive Schedule (LAS).

## 6 Experiments

We evaluate the proposed discretization schedule on both synthetic toy distributions and a large-scale image generation benchmark.

### 6.1 Toy Examples: Gaussian Mixture Models

We first consider controlled synthetic settings where the ground-truth data distribution is a Gaussian mixture model (GMM). The corresponding details are provided in Appendix H.

### 6.2 ImageNet $256 \times 256$ with Latent Diffusion

We next evaluate on a real-world generative modeling task using latent diffusion models on ImageNet $256 \times 256$ (Rombach et al., 2022).

We use the class-conditional ImageNet LDM-VQ-8 model. Its exact discrete support and entropy upper bound are detailed in Remark 4.4.

We use classifier guidance with scale 2 and report Fréchet Inception Distance (FID). We test two samplers: (i) DDIM with $\eta = 1$, which corresponds to the stochastic sampler consistent with our "freezing-$m$" discretization, and (ii) SDE-DPM-Solver++(2M) and DPM-Solver++(2M) second-order samplers (Lu et al., 2025).

For SDE-DPM-Solver++(2M) and DPM-Solver++(2M), we found the method to be sensitive to highly inhomogeneous step sizes due to its second-order structure. In particular, second-order solvers are most stable when consecutive step sizes are *comparable* (e.g., nearly constant on the log-SNR axis): their local truncation error analysis and practical error cancellation across adjacent steps can break down when the schedule is highly inhomogeneous, leading to instability and degraded sample quality. To stabilize second-order sampling, we therefore encourage *smooth* log-SNR step sizes by adding an additional smoothness penalty to the schedule optimization objective:

$$\alpha \sum_{k=2}^{K} (h_k - h_{k-1})^2, \tag{20}$$

where $h_k := \log(\gamma_k / \gamma_{k-1})$ denotes the log-SNR ratio at step $k$.

**Schedules and hyperparameters** We select $(\lambda, \alpha)$ using a small pilot budget of 1,000 generated samples and fix them thereafter. All numbers reported in Table 1 are computed from an independent run of 50,000 generated samples using the fixed hyperparameters. This mirrors standard practice for sampler hyperparameter tuning and does not reuse the evaluation budget during selection. The comparison schedules are standard fixed heuristics; unlike LAS, they do not use the trained model's loss profile to adapt the allocation of sampling steps. Throughout all experiments we set the SNR-axis regularization parameter to $\lambda = 1.5$. Also, we use $\alpha = 12$ for all DPM-Solver experiments. Additional ablations and stability checks are deferred to Appendix I. There, we study the sensitivity of LAS to its regularization parameters, compare against Align Your Steps (AYS), and examine mixed schedules that optimize only part of the reverse trajectory. These experiments indicate that LAS is stable across a range of hyperparameters and that the KL-based surrogate is most informative away from the near-terminal high-SNR regime.

We compare the proposed LAS with three commonly used time-discretization schedules: Time-uniform, LogSNR, and EDM (Karras et al., 2022).

Table 1 reports results on ImageNet $256 \times 256$ for number of function evaluations (NFE) 10 and 20. Overall, LAS improves FID over the time-uniform schedule for all three samplers in the reported settings and is competitive across the other metrics. The gains are particularly pronounced for the first-order method DDIM at low NFE (e.g., NFE= 10), which is consistent with the theory suggesting discretization effects are most visible in coarse discretizations. Thus LAS can speed up generation by reaching a target sample quality with fewer function evaluations, while remaining complementary to one-step or distillation-based methods. We also observe improvements for SDE-DPM++(2M) and DPM-Solver++ (2M).

A direct comparison with Align Your Steps (AYS) is provided in Appendix I. LAS is competitive with AYS while requiring substantially lighter post-training schedule optimization: LAS uses training-loss information and dynamic programming, whereas AYS requires a separate Monte Carlo-based schedule-search procedure.

## 7 Scope and Limitations

Our theoretical discretization bound is stated for discrete or countable target distributions with finite order-1/2 Rényi entropy. This setting is not only a mathematical abstraction: it directly covers generative models whose continuous states are obtained from discrete latent codes. In particular, VQ-based latent diffusion models first choose codebook indices from a finite vocabulary and then map them to Euclidean codebook embeddings; the resulting latent variable is therefore supported on a finite, hence countable, subset of Euclidean space. This includes the VQ-latent ImageNet latent-diffusion (Rombach et al., 2022) setting considered in our experiments. The same viewpoint also applies to language modeling settings in which a discrete token sequence is embedded into Euclidean space and then modeled by a Gaussian diffusion process over embeddings.

The countable-support assumption is used specifically for the entropy-based MMSE derivative bound in Theorem 4.8. It is not needed for the KL decomposition, the exact MMSE representation of $\mathcal{E}_{\text{disc}}$, the loss identity underlying LAS, or the LAS algorithm itself, which require only the corresponding moment and integrability conditions. Our proof of Theorem 4.8 uses posterior mass functions and sums over the support to control posterior moments through $H_{1/2}$; extending this particular entropy bound to general continuous target distributions is therefore not a direct replacement of sums by integrals. Such an extension would likely require quantization arguments or additional regularity assumptions involving continuous information quantities, and we leave it as an important direction for future work.

The assumption is used to obtain an entropy-controlled bound on the derivative of the Gaussian-channel MMSE. The bound should therefore be interpreted as replacing explicit ambient-dimensional dependence by information-theoretic dependence. The complexity of the bound is captured by quantities such as $H_{1/2}$, which may itself scale with the effective number of latent codes or tokens.

## 8 Future work

Our discretization bound is obtained by controlling the MMSE derivative via a general upper bound, and we expect this step is not tight in many regimes. In particular, under additional but still reasonable assumptions on the code distribution (e.g., separation properties of the code support), the MMSE regularity may admit sharper control, leading to improved constants or rates beyond the current $O(H_{1/2}^2/K)$ dependence. More broadly, we believe that exploiting local structure of the code distribution could yield sharper convergence guarantees.

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

## A  Proof of Proposition 3.1

We adapt the argument of Theorem 10 in Chen et al. (2023b).

Let

$$\mathcal{C} := C([0, T_\delta]; \mathbb{R}^d)$$

be equipped with its Borel $\sigma$-field and canonical filtration.

For $t \in (s_{k-1}, s_k]$ and $\omega \in \mathcal{C}$, define the canonical drift mismatch

$$\Delta_t(\omega) = \frac{m_{T-t}\big(\omega(t)\big) - \widehat{m}_{T-s_{k-1}}\big(\omega(s_{k-1})\big)}{T - t}. \tag{21}$$

We assume, as is standard for a learned denoiser, that each map

$$\widehat{m}_{T-s_{k-1}} : \mathbb{R}^d \to \mathbb{R}^d$$

is finite-valued and Borel measurable. Define

$$A_t(\omega) := \int_0^t \|\Delta_s(\omega)\|_2^2 \, ds$$

and the canonical stopping-time functional

$$\tau_n(\omega) := \inf\big\{t \in [0, T_\delta] : A_t(\omega) \geq n\big\} \wedge T_\delta.$$

We first record that

$$A_{T_\delta}(\omega) < \infty \qquad \text{for every } \omega \in \mathcal{C}. \tag{22}$$

Indeed, writing $\varphi_t$ for the centered Gaussian density with covariance $tI_d$, the true denoiser admits the representation

$$m_t(x) = \frac{\displaystyle\int_{\mathbb{R}^d} z \, \varphi_t(x - z) \, p(dz)}{\displaystyle\int_{\mathbb{R}^d} \varphi_t(x - z) \, p(dz)}.$$

Under the finite-second-moment assumption, the numerator is jointly continuous in $(t, x)$ on $(0, \infty) \times \mathbb{R}^d$. The denominator is jointly continuous and strictly positive. Consequently,

$$(t, x) \longmapsto m_t(x)$$

is jointly continuous, and in particular jointly locally bounded, on $[\delta, T] \times \mathbb{R}^d$.

Fix $\omega \in \mathcal{C}$ and set

$$R_\omega := \sup_{0 \leq t \leq T_\delta} \|\omega(t)\|_2.$$

Then

$$M_\omega := \sup_{\substack{u \in [\delta, T] \\ \|x\|_2 \leq R_\omega}} \|m_u(x)\|_2 < \infty.$$

Since the grid has only finitely many points and every learned-denoiser value is finite,

$$\widehat{M}_\omega := \max_{1 \leq k \leq K} \left\| \widehat{m}_{T-s_{k-1}} \big( \omega(s_{k-1}) \big) \right\|_2 < \infty.$$

Because $T - t \geq \delta$ on $[0, T_\delta]$, equation (21) gives

$$\|\Delta_t(\omega)\|_2 \leq \frac{M_\omega + \widehat{M}_\omega}{\delta}.$$

Hence

$$A_{T_\delta}(\omega) \leq \frac{T_\delta}{\delta^2} \big( M_\omega + \widehat{M}_\omega \big)^2 < \infty,$$

which proves (22). In particular,

$$\tau_n(\omega) = T_\delta \quad \text{for every } n > A_{T_\delta}(\omega). \tag{23}$$

Under $\mathbb{P}$,

$$dY_t = \beta_t(Y_t)\, dt + dB_t,$$

where $B_t$ is a $\mathbb{P}$-Brownian motion. For $n \in \mathbb{N}$, define

$$\mathcal{E}_t^{(n)} := \exp \left( - \int_0^{t \wedge \tau_n(Y)} \Delta_s(Y)^\top \, dB_s - \frac{1}{2} \int_0^{t \wedge \tau_n(Y)} \|\Delta_s(Y)\|_2^2 \, ds \right).$$

By definition of $\tau_n$,

$$\int_0^{\tau_n(Y)} \|\Delta_s(Y)\|_2^2 \, ds \leq n \qquad \mathbb{P}\text{-a.s.}$$

Therefore Novikov's condition holds for the stopped integrand, and $(\mathcal{E}_t^{(n)})_{t \leq T_\delta}$ is a true $\mathbb{P}$-martingale.

Define a probability measure $\widetilde{\mathbb{P}}_n$ on $\mathcal{C}$ by

$$\frac{d\widetilde{\mathbb{P}}_n}{d\mathbb{P}} := \mathcal{E}_{T_\delta}^{(n)}.$$

By Girsanov's theorem,

$$B_t^{(n)} := B_t + \int_0^{t \wedge \tau_n(Y)} \Delta_s(Y) \, ds$$

is a $\widetilde{\mathbb{P}}_n$-Brownian motion. Hence, under $\widetilde{\mathbb{P}}_n$,

$$dY_t = \widetilde{\beta}_t(Y_t) 1_{\{t \leq \tau_n(Y)\}} \, dt + \beta_t(Y_t) 1_{\{t > \tau_n(Y)\}} \, dt + dB_t^{(n)}. \tag{24}$$

Moreover,

$$\begin{aligned}
\mathrm{KL}(\mathbb{P} \| \widetilde{\mathbb{P}}_n) &= \mathbb{E}_\mathbb{P} \left[ - \log \mathcal{E}_{T_\delta}^{(n)} \right] \\
&= \mathbb{E}_\mathbb{P} \left[ \int_0^{\tau_n(Y)} \Delta_s(Y)^\top \, dB_s + \frac{1}{2} \int_0^{\tau_n(Y)} \|\Delta_s(Y)\|_2^2 \, ds \right].
\end{aligned}$$

The stopped stochastic integral is square-integrable, since its quadratic variation is bounded by $n$, and therefore has mean zero. Thus

$$\mathrm{KL}(\mathbb{P} \| \widetilde{\mathbb{P}}_n) = \frac{1}{2} \mathbb{E}_\mathbb{P} \left[ \int_0^{\tau_n(Y)} \|\Delta_s(Y)\|_2^2 \, ds \right] \leq \frac{1}{2} \mathbb{E}_\mathbb{P} \left[ \int_0^{T_\delta} \|\Delta_s(Y)\|_2^2 \, ds \right]. \tag{25}$$

Let $\overline{\mathbb{P}}$ denote the path law of the approximate reverse dynamics with the correct initial law:

$$d\overline{Y}_s = \widetilde{\beta}_s(\overline{Y}_s)\, ds + dB_s, \qquad \overline{Y}_0 \sim p_T.$$

It remains to identify the limit of $\widetilde{\mathbb{P}}_n$. Let $\widetilde{\xi}$ be a realization of the approximate process on another probability space, driven by a Brownian motion $\overline{B}$, so that

$$d\widetilde{\xi}_t = \widetilde{\beta}_t(\widetilde{\xi}_t)\,dt + d\overline{B}_t, \qquad \mathcal{L}(\widetilde{\xi}) = \overline{\mathbb{P}}.$$

For every $n$, define

$$\sigma_n := \tau_n(\widetilde{\xi}).$$

Construct $\widetilde{\xi}^n$ using the same initial condition and Brownian motion by setting

$$\widetilde{\xi}_t^n = \widetilde{\xi}_t, \qquad 0 \le t \le \sigma_n,$$

and, conditionally on $\widetilde{\xi}_{\sigma_n}^n = \widetilde{\xi}_{\sigma_n}$, continuing after $\sigma_n$ according to

$$d\widetilde{\xi}_t^n = \beta_t(\widetilde{\xi}_t^n)\,dt + d\overline{B}_t, \qquad t > \sigma_n.$$

For completeness, these equations are well posed. The approximate equation is pathwise unique by solving successively on each grid interval, where it is a linear SDE in the current state. The exact drift is smooth and locally Lipschitz on $[0, T_\delta] \times \mathbb{R}^d$. Its nonexplosive reverse transition law from $(r, x)$ is the Doob $h$-transform with transition density

$$q_{r,t}(x, y) = \frac{\varphi_{t-r}(y - x)\, p_{T-t}(y)}{p_{T-r}(x)}, \qquad 0 \le r < t \le T_\delta.$$

Thus the exact equation, and consequently the concatenated switched equation, is unique in law.

Since $\widetilde{\xi}^n$ and $\widetilde{\xi}$ agree up to $\sigma_n$, and $\tau_n$ is nonanticipative,

$$\tau_n(\widetilde{\xi}^n) = \tau_n(\widetilde{\xi}) = \sigma_n.$$

It follows that $\widetilde{\xi}^n$ satisfies

$$d\widetilde{\xi}_t^n = \widetilde{\beta}_t(\widetilde{\xi}_t^n)1_{\{t \le \tau_n(\widetilde{\xi}^n)\}}\,dt + \beta_t(\widetilde{\xi}_t^n)1_{\{t > \tau_n(\widetilde{\xi}^n)\}}\,dt + d\overline{B}_t.$$

Comparison with (24) and uniqueness in law therefore gives

$$\mathcal{L}(\widetilde{\xi}^n) = \widetilde{\mathbb{P}}_n.$$

By (22),

$$A_{T_\delta}(\widetilde{\xi}) < \infty \qquad \text{almost surely.}$$

Consequently, by (23),

$$\sigma_n = T_\delta \qquad \text{for all sufficiently large } n, \quad \text{almost surely.}$$

In particular,

$$\widetilde{\xi}^n \longrightarrow \widetilde{\xi} \qquad \text{almost surely in } \mathcal{C},$$

and hence

$$\widetilde{\mathbb{P}}_n \Rightarrow \overline{\mathbb{P}}. \tag{26}$$

Relative entropy is jointly lower semicontinuous under weak convergence on the Polish space $\mathcal{C}$. Since the first argument is fixed, (26) and (25) yield

$$\mathrm{KL}(\mathbb{P}\|\overline{\mathbb{P}}) \le \liminf_{n \to \infty} \mathrm{KL}(\mathbb{P}\|\widetilde{\mathbb{P}}_n)$$

$$\le \frac{1}{2}\mathbb{E}_{\mathbb{P}}\left[\int_0^{T_\delta} \|\Delta_s(Y)\|_2^2\,ds\right]$$

$$= \frac{1}{2}\mathbb{E}_{\mathbb{P}}\left[\int_0^{T_\delta} \left\|\beta_s(Y_s) - \widetilde{\beta}_s(Y)\right\|_2^2\,ds\right]$$

Finally, note that $\overline{\mathbb{P}}$ and $\widetilde{\mathbb{P}}$ have the same drift and differ only in their initial distributions. Let

$$e_0 : \mathcal{C} \to \mathbb{R}^d, \qquad e_0(\gamma) = \gamma(0).$$

Because $\overline{\mathbb{P}}$ and $\widetilde{\mathbb{P}}$ have identical conditional path laws given the initial state,

$$\frac{d\overline{\mathbb{P}}}{d\widetilde{\mathbb{P}}}(\gamma) = \frac{dp_T}{dq}\big(e_0(\gamma)\big).$$

Consequently, $\tilde{\mathbb{P}}$-almost surely, and hence $\mathbb{P}$-almost surely,

$$\log \frac{d\mathbb{P}}{d\widetilde{\mathbb{P}}} = \log \frac{d\mathbb{P}}{d\overline{\mathbb{P}}} + \log \frac{dp_T}{dq}(\gamma(0)).$$

Taking expectation under $\mathbb{P}$ yields

$$
\begin{aligned}
\mathrm{KL}\Big(\mathbb{P}\,\Big\|\,\widetilde{\mathbb{P}}\Big) &= \mathrm{KL}\big(\mathbb{P}\,\big\|\,\overline{\mathbb{P}}\big) + \mathbb{E}_{\gamma(0)\sim p_T}\left[\log \frac{dp_T}{dq}(\gamma(0))\right] \\
&= \mathrm{KL}\big(\mathbb{P}\,\big\|\,\overline{\mathbb{P}}\big) + \mathrm{KL}(p_T\|q) \\
&\le \mathrm{KL}(p_T\|q) + \frac{1}{2}\,\mathbb{E}_{\mathbb{P}}\left[\int_0^{T_\delta} \|\delta_s\|^2\, ds\right].
\end{aligned}
$$

## B  Proof of Proposition 4.6

*Proof.* The forward process is $X_t = Z + W_t$ for $t \in [0, T]$, where $Z \sim p$ is independent of the standard Brownian motion $(W_t)_{t\ge 0}$. Parameterize observations by the signal-to-noise ratio $\gamma = 1/t > 0$, so the observation at SNR $\gamma$ is $X_{1/\gamma} = Z + W_{1/\gamma}$.

Define the increasing filtration by

$$\mathcal{F}_\gamma := \sigma\big(X_{1/\gamma'} : \frac{1}{T} \le \gamma' \le \gamma\big), \qquad \gamma \in \left[\frac{1}{T}, \frac{1}{\delta}\right].$$

Higher $\gamma$ corresponds to lower noise variance $1/\gamma$, so the filtration is increasing: $\gamma_1 < \gamma_2$ implies $\mathcal{F}_{\gamma_1} \subset \mathcal{F}_{\gamma_2}$.

Let

$$M_\gamma := \mathbb{E}[Z \mid \mathcal{F}_\gamma] = \mathbb{E}\big[Z \mid X_{1/\gamma}\big] = m_{1/\gamma}(X_{1/\gamma})$$

be the posterior mean (Bayes-optimal denoiser) at SNR $\gamma$.

By the tower property, for $\gamma_1 < \gamma_2$,

$$M_{\gamma_1} = \mathbb{E}\big[Z \mid \mathcal{F}_{\gamma_1}\big] = \mathbb{E}\big[\mathbb{E}[Z \mid \mathcal{F}_{\gamma_2}] \mid \mathcal{F}_{\gamma_1}\big] = \mathbb{E}[M_{\gamma_2} \mid \mathcal{F}_{\gamma_1}].$$

Thus, $(M_\gamma)_{\gamma>0}$ is a vector-valued martingale with respect to $(\mathcal{F}_\gamma)$—the Doob martingale associated with the integrable target $Z$.

The MMSE at SNR $\gamma$ is

$$\mathrm{mmse}(\gamma) := \mathbb{E}\big[\|Z - M_\gamma\|_2^2\big].$$

Since $\mathcal{F}_{\gamma_{k-1}} \subset \mathcal{F}_\gamma$ for $\gamma > \gamma_{k-1}$, the corresponding subspaces of $\mathcal{F}_\gamma$-measurable random vectors are nested. The error $Z - M_\gamma$ is orthogonal (in $L^2(\mathbb{P})$) to all $\mathcal{F}_\gamma$-measurable functions, and in particular to the martingale increment $M_\gamma - M_{\gamma_{k-1}}$.

Decompose

$$Z - M_{\gamma_{k-1}} = (Z - M_\gamma) + (M_\gamma - M_{\gamma_{k-1}}).$$

The cross-term vanishes:

$$\mathbb{E}\big[(Z - M_\gamma)^\top (M_\gamma - M_{\gamma_{k-1}})\big] = 0,$$

so by the Pythagorean theorem,

$$\mathbb{E}\big[\|Z - M_{\gamma_{k-1}}\|_2^2\big] = \mathbb{E}\big[\|Z - M_\gamma\|_2^2\big] + \mathbb{E}\big[\|M_\gamma - M_{\gamma_{k-1}}\|_2^2\big].$$

Hence,

$$\mathbb{E}\big[\|M_\gamma - M_{\gamma_{k-1}}\|_2^2\big] = \mathrm{mmse}(\gamma_{k-1}) - \mathrm{mmse}(\gamma). \tag{27}$$

Now express $\mathcal{E}_{\mathrm{disc}}$. By definition,

$$\mathcal{E}_{\mathrm{disc}} = \sum_{k=1}^K \mathbb{E}\left[\int_{s_{k-1}}^{s_k} \|m_{T-s}(Y_s) - m_{T-s_{k-1}}(Y_{s_{k-1}})\|_2^2 \frac{ds}{(T-s)^2}\right].$$

The change of variables $\gamma = 1/(T-s)$ gives $d\gamma = ds/(T-s)^2$, and as $s$ runs from $s_{k-1}$ to $s_k$, $\gamma$ runs from $\gamma_{k-1}$ to $\gamma_k$ (increasing). Since we also have $(Y_s)_{s\in[0,T_\delta]} = (X_{T-s})_{s\in[0,T_\delta]}$, each summand becomes

$$\mathbb{E}\left[\int_{s_{k-1}}^{s_k} \|m_{T-s}(Y_s) - m_{T-s_{k-1}}(Y_{s_{k-1}})\|_2^2 \frac{ds}{(T-s)^2}\right] = \mathbb{E}\left[\int_{\gamma_{k-1}}^{\gamma_k} \|M_\gamma - M_{\gamma_{k-1}}\|_2^2 \, d\gamma\right].$$

Interchanging the order of expectation and integration (justified by Tonelli's theorem),

$$\mathbb{E}\left[\int_{\gamma_{k-1}}^{\gamma_k} \|M_\gamma - M_{\gamma_{k-1}}\|_2^2 \, d\gamma\right] = \int_{\gamma_{k-1}}^{\gamma_k} \mathbb{E}\big[\|M_\gamma - M_{\gamma_{k-1}}\|_2^2\big] \, d\gamma.$$

Applying (27), this becomes

$$\int_{\gamma_{k-1}}^{\gamma_k} \big(\mathrm{mmse}(\gamma_{k-1}) - \mathrm{mmse}(\gamma)\big) \, d\gamma.$$

Summing over $k = 1, \ldots, K$ yields the desired expression for $\mathcal{E}_{\mathrm{disc}}$.

This expresses the discretization error as the total expected quadratic variation of the missed martingale increments when the denoiser is held constant within each reverse step, instead of following the continuous Doob martingale $M_\gamma$. $\qquad\square$

## C  Proof of Theorem 4.8 and Corollary 4.9

We now show how the derivative bound on the MMSE in Theorem 4.8 is obtained. The argument proceeds in three steps. First, we express the derivative of the MMSE in terms of a conditional covariance. Second, we control the trace of the squared covariance by a fourth-moment quantity. Finally, we bound this fourth moment using Rényi entropy of the target distribution.

Fix $t \in [0, T]$. According to the forward process, the joint law of $(Z, X_t)$ is

$$\mathbb{P}\big(Z = z, \ X_t \in dx\big) = p(z)\, p_t(x \mid z)\, dx, \qquad z \in \mathcal{C},$$

where $p_t(x \mid z)$ is the probability density function of Gaussian distribution $\mathcal{N}(z, tI_d)$. We compute the posterior

$$p_t^{\mathrm{post}}(z \mid x) = \frac{p(z)p_t(x \mid z)}{p_{X_t}(x)}, \tag{28}$$

where $p_{X_t}(x) = \sum_{u\in\mathcal{C}} p(u)p_t(x \mid u)$ is the probability density function of $X_t$.

Introduce a new random variable $Z'$. We specify the joint law of $(Z, X_t, Z')$ by

$$\mathbb{P}\big(Z = z, \ X_t \in dx, \ Z' = z'\big) = p(z)\, p_t(x \mid z)\, r_t(z' \mid x, z)\, dx, \qquad z, z' \in \mathcal{C}.$$

where $r_t(z' \mid x, z) = p_t^{\mathrm{post}}(z' \mid x)$.

Note that conditional on $X_t$, the random variable $Z'$ can be considered as a posterior draw independent of $Z$. Indeed,

$$
\begin{aligned}
p_{Z,Z'|X_t}(z, z' \mid x) &= \frac{p(z)\, p_t(x \mid z)\, r_t(z' \mid x, z)}{p_{X_t}(x)} \\
&= \frac{p(z)\, p_t(x \mid z)}{p_{X_t}(x)}\, p_t^{\mathrm{post}}(z' \mid x) \\
&= p_t^{\mathrm{post}}(z \mid x)\, p_t^{\mathrm{post}}(z' \mid x).
\end{aligned}
$$

Define the kernel

$$
q_{t,z}(z') := \mathbb{P}(Z' = z' \mid Z = z) = \int_{\mathbb{R}^d} p_t^{\mathrm{post}}(z' \mid x)\, p_t(x \mid z)\, dx, \qquad z, z' \in \mathcal{C}.
$$

The proof for Theorem 4.8 starts from a recent identity from information theory, which relates the derivative of the MMSE along the Gaussian channel to the conditional covariance of the posterior. The following proposition is a reformulation of a result from Nguyen (2024).

**Proposition C.1.** *The MMSE function*

$$
\mathrm{mmse}(\gamma) = \mathbb{E}\big[\|Z - \mathbb{E}[Z \mid X_{1/\gamma}]\|_2^2\big]
$$

*is differentiable and monotonically decreasing for $\gamma > 0$, and*

$$
|\mathrm{mmse}'(\gamma)| = -\mathrm{mmse}'(\gamma) = \mathbb{E}\big[\mathrm{tr}\big(\mathrm{Cov}(Z \mid X_{1/\gamma})^2\big)\big] = \mathbb{E}\big[\mathrm{tr}\big(\mathrm{Cov}(Z \mid X_t)^2\big)\big], \tag{29}
$$

*where $t = 1/\gamma$.*

*Proof.* Recall we defined the reverse-time process $(Y_s)_{s \in [0,T]}$ by $Y_s := X_{T-s}$ for all $s \in [0,T]$. Let $\mathcal{F}_s := \sigma(Y_u : 0 \le u \le s)$ be the natural filtration of $(Y_s)_{s \in [0,T]}$. Define

$$
M_s := \mathbb{E}[Y_T \mid \mathcal{F}_s].
$$

Since $Y$ is Markov, $\mathbb{E}[Y_T \mid \mathcal{F}_s] = \mathbb{E}[Y_T \mid Y_s]$, hence $M_s = u(s, Y_s)$ where $u(s, y) := \mathbb{E}[Y_T \mid Y_s = y]$. For $s \in [0, T_\delta]$, we have

$$
u(s, y) = m_{T-s}(y) = y + (T - s)\nabla \log p_{T-s}(y).
$$

Since $p_t = p * \mathcal{N}(0, tI_d)$ is strictly positive and smooth for every $t > 0$, and $T - s \ge \delta$, it follows that $u \in C^{1,2}([0, T_\delta] \times \mathbb{R}^d)$, which justifies the following application of Itô's formula to $u$.

Since $(M_s)_{s \in [0,T]}$ is a square-integrable continuous martingale, and moreover, the diffusion coefficient of the reverse SDE is the identity, we have

$$
dM_s = \nabla_y u(s, Y_s)\, dB_s
$$

by Itô's formula. Since $M_s$ is a continuous martingale, Itô's formula gives

$$
d\|M_s\|_2^2 = 2\langle M_s, dM_s \rangle + d\langle M \rangle_s,
$$

and taking expectation kills the martingale term, hence

$$
\frac{d}{ds}\mathbb{E}\|M_s\|_2^2 = \mathbb{E}\left[\frac{d}{ds}\langle M \rangle_s\right] = \mathbb{E}\big[\|\nabla_y u(s, Y_s)\|_F^2\big],
$$

where $\|\cdot\|_F$ is the Frobenius norm.

Define

$$
\mathrm{mmse}_{\mathrm{rev}}(s) := \mathbb{E}\big[\|Y_T - \mathbb{E}[Y_T \mid \mathcal{F}_s]\|_2^2\big] = \mathbb{E}\big[\|Y_T - M_s\|_2^2\big].
$$

Using orthogonality of conditional expectation,

$$\mathrm{mmse}_{\mathrm{rev}}(s) = \mathbb{E}\|Y_T\|_2^2 - \mathbb{E}\|M_s\|_2^2.$$

Differentiating in $s$ gives

$$\mathrm{mmse}_{\mathrm{rev}}'(s) = -\mathbb{E}\big[\|\nabla_y u(s, Y_s)\|_F^2\big]. \tag{30}$$

Recall that $Y_s = X_{T-s}$ and $Y_T = Z$. Thus, $u(s,y) = \mathbb{E}[Z \mid X_{T-s} = y]$. Denote $m_t(y) := \mathbb{E}[Z \mid X_t = y]$ and $\Sigma_t(y) := \mathrm{Cov}(Z \mid X_t = y)$. A standard differentiation-under-the-integral calculation for the Gaussian channel (see e.g. Tweedie-type identities) yields the matrix Jacobian identity

$$\nabla_y m_t(y) = \frac{1}{t}\Sigma_t(y). \tag{31}$$

(Quick derivation: $m_t(y) = \frac{\int z\, p_Z(z)\varphi_t(y-z)\, dz}{\int p_Z(z)\varphi_t(y-z)\, dz}$, differentiate using $\nabla_y \varphi_t(y - z) = -(y - z)\varphi_t(y - z)/t$, and simplify to obtain $\nabla_y m_t(y) = \frac{1}{t}\big(\mathbb{E}[ZZ^\top \mid X_t = y] - m_t(y)m_t(y)^\top\big)$.)

Combining (30)–(31) and using $\|\Sigma\|_F^2 = \mathrm{tr}(\Sigma^2)$ for symmetric $\Sigma$,

$$\mathrm{mmse}_{\mathrm{rev}}'(s) = -\mathbb{E}\left[\left\|\frac{1}{t}\Sigma_t(X_t)\right\|_F^2\right] = -\frac{1}{t^2}\mathbb{E}\big[\mathrm{tr}\big(\mathrm{Cov}(Z \mid X_t)^2\big)\big],$$

where $t = T - s$.

Note that since $Y_s = X_t$ and $Y_T = Z$, we have $\mathrm{mmse}(\gamma) = \mathrm{mmse}_{\mathrm{rev}}(s)$, where $\gamma = 1/t = 1/(T - s)$. By the chain rule,

$$\mathrm{mmse}'(\gamma) = \mathrm{mmse}_{\mathrm{rev}}'(s) \cdot \frac{ds}{d\gamma} = \big(-\gamma^2\, \mathbb{E}\big[\mathrm{tr}\big(\mathrm{Cov}(Z \mid X_{1/\gamma})^2\big)\big]\big) \cdot \left(\frac{1}{\gamma^2}\right),$$

hence (29). In particular, $\mathrm{mmse}'(\gamma) \leq 0$ and mmse is monotonically decreasing. $\qquad\square$

Identity (29) reduces the problem of bounding $\mathrm{mmse}'(\gamma)$ to controlling the squared conditional covariance of the posterior distribution of $Z$ given a noisy observation.

To control the right-hand side of (29), we bound the trace of the squared covariance matrix by a fourth moment using the following probabilistic lemma.

**Lemma C.2.** *Let $Y$ be an $\mathbb{R}^d$-valued random vector with mean $m := \mathbb{E}[Y]$ and covariance $\Sigma := \mathrm{Cov}(Y)$. Then for any $a \in \mathbb{R}^d$,*

$$\mathrm{tr}(\Sigma^2) \leq \mathbb{E}\big[\|Y - a\|_2^4\big].$$

*Proof.* Since the covariance matrix $\Sigma$ is symmetric positive semidefinite, its eigenvalues $\lambda_1, \ldots, \lambda_d$ are all nonnegative. This implies

$$\mathrm{tr}(\Sigma^2) = \sum_{i=1}^d \lambda_i^2 \ \leq \ \left(\sum_{i=1}^d \lambda_i\right)^2 = (\mathrm{tr}\,\Sigma)^2.$$

Then we evaluate

$$\mathrm{tr}\,\Sigma = \mathrm{tr}\big(\mathbb{E}[(Y - m)(Y - m)^\top]\big) = \mathbb{E}\,\mathrm{tr}\big((Y - m)(Y - m)^\top\big) = \mathbb{E}\|Y - m\|_2^2,$$

where we used linearity of trace and expectation, and the identity $\mathrm{tr}(vv^\top) = \|v\|_2^2$ for any vector $v$.

Since $m = \mathbb{E}Y$ is the unique minimizer of the function $a \mapsto \mathbb{E}\|Y - a\|_2^2$,

$$\mathbb{E}\|Y - m\|_2^2 \ \leq \ \mathbb{E}\|Y - a\|_2^2 \qquad \text{for every } a \in \mathbb{R}^d.$$

Combining these and finally applying the Cauchy–Schwarz inequality yields for every $a \in \mathbb{R}^d$,

$$\mathrm{tr}(\Sigma^2) \ \leq \ (\mathrm{tr}\,\Sigma)^2 \ = \ \big(\mathbb{E}\|Y - m\|_2^2\big)^2 \ \leq \ \big(\mathbb{E}\|Y - a\|_2^2\big)^2 \ \leq \ \mathbb{E}\|Y - a\|_2^4.$$

$\qquad\square$

Applying Lemma C.2 yields the following bound for the trace of the squared covariance matrix.

**Proposition C.3.** *For each $z \in \mathcal{C}$,*

$$\mathbb{E}\big[\mathrm{tr}\big(\mathrm{Cov}(Z' \mid X_t)^2\big) \mid Z = z\big] \leq \mathbb{E}_{Z' \sim q_{t,z}}\big[\|Z' - z\|_2^4\big],$$

*where $q_{t,z}(z') := \int p_t^{\mathrm{post}}(z' \mid x)\, p_t(x \mid z)\, dx$.*

*Moreover,*

$$\mathbb{E}\big[\mathrm{tr}\big(\mathrm{Cov}(Z' \mid X_t)^2\big)\big] \leq \mathbb{E}\big[\|Z' - Z\|_2^4\big].$$

*Proof.* For every $z \in \mathcal{C}$, we have

$$\mathbb{E}\big[\mathrm{tr}\big(\mathrm{Cov}(Z' \mid X_t)^2\big) \mid Z = z\big] = \int \mathrm{tr}\big(\mathrm{Cov}_{\xi \sim p_t^{\mathrm{post}}(\cdot \mid x)}(\xi)^2\big)\, p_t(x \mid z)\, dx$$

$$\leq \int \mathbb{E}_{\xi \sim p_t^{\mathrm{post}}(\cdot \mid x)}\big[\|\xi - z\|_2^4\big]\, p_t(x \mid z)\, dx \qquad (32)$$

$$= \int \int \|\xi - z\|_2^4\, p_t^{\mathrm{post}}(\xi \mid x)\, p_t(x \mid z)\, d\xi\, dx$$

$$= \int \int \|\xi - z\|_2^4\, q_{t,z}(\xi)\, d\xi \qquad (33)$$

$$= \mathbb{E}_{Z' \sim q_{t,z}}\big[\|Z' - z\|_2^4\big],$$

where for (32), we applied Lemma C.2 with anchor point $a = z$ (which is constant given the outer conditioning on $Z = z$) to the conditional distributions; and (33) follows from the Tonelli's theorem.

Finally, taking expectation over $Z \sim p(\cdot)$ and using the definition $q_{t,z}(z') = \mathbb{P}(Z' = z' \mid Z = z)$ gives

$$\mathbb{E}\big[\mathrm{tr}\big(\mathrm{Cov}(Z' \mid X_t)^2\big)\big] = \sum_{z \in \mathcal{C}} p(z)\, \mathbb{E}\big[\mathrm{tr}\big(\mathrm{Cov}(Z' \mid X_t)^2\big) \,\big|\, Z = z\big]$$

$$\leq \sum_{z \in \mathcal{C}} p(z)\, \mathbb{E}_{Z' \sim q_{t,z}}\big[\|Z' - z\|_2^4\big]$$

$$= \mathbb{E}_Z\Big[\mathbb{E}\big[\|Z' - Z\|_2^4 \,\big|\, Z\big]\Big]$$

$$= \mathbb{E}\big[\|Z' - Z\|_2^4\big].$$

$$\square$$

Combined with Proposition C.1, this shows that controlling $|\mathrm{mmse}'(\gamma)|$ reduces to bounding a fourth moment of the posterior fluctuations.

The final step is to bound $\mathbb{E}[\|Z' - Z\|_2^4]$ using Rényi entropy of order $1/2$, an information-theoretic quantity of the target distribution.

**Proposition C.4.** *For all $t > 0$,*

$$\mathbb{E}\big[\|Z' - Z\|_2^4\big] \leq 96t^2 H_{1/2}^2 + 64t^2\Big(H_{1/2} + 1\Big). \qquad (34)$$

*Proof.* The inequality is trivial for $H_{1/2} = \infty$. We only need to prove for $H_{1/2} < \infty$.

For any $z' \neq z$, keeping only two terms in the denominator of (28) gives

$$p_t^{\mathrm{post}}(z' \mid x) \leq \frac{p(z')p_t(x \mid z')}{p(z')p_t(x \mid z') + p(z)p_t(x \mid z)}.$$

Apply the AM–GM inequality to the denominator yields

$$p(z')p_t(x \mid z') + p(z)p_t(x \mid z) \geq 2\sqrt{p(z')p_t(x \mid z')p(z)p_t(x \mid z)}.$$

Hence,

$$p_t^{\text{post}}(z' \mid x) \le \frac{p(z')p_t(x \mid z')}{2\sqrt{p(z')p_t(x \mid z')p(z)p_t(x \mid z)}} = \frac{1}{2}\sqrt{\frac{p(z')p_t(x \mid z')}{p(z)p_t(x \mid z)}},$$

and

$$q_{t,z}(z') = \int p_t^{\text{post}}(z' \mid x)\, p_t(x \mid z)\, dx \le \tfrac{1}{2}\sqrt{\frac{p(z')}{p(z)}} \int \sqrt{p_t(x \mid z')p_t(x \mid z)}\, dx.$$

For Gaussian kernels $p_t(x \mid z)$ and $p_t(x \mid z')$, one computes

$$\int \sqrt{p_t(x \mid z')p_t(x \mid z)}\, dx = \exp\left(-\frac{\|z' - z\|_2^2}{8t}\right).$$

Hence,

$$q_{t,z}(z') \le \tfrac{1}{2}\sqrt{\frac{p(z')}{p(z)}} \exp\left(-\frac{\|z' - z\|_2^2}{8t}\right).$$

Denote $R := \|Z' - Z\|_2$. For any $r \ge 0$,

$$
\begin{aligned}
\mathbb{P}(R > r \mid Z = z) &= \sum_{\|z'-z\|_2 > r} q_{t,z}(z') \\
&\le \frac{1}{2\sqrt{p(z)}} \sum_{\|z'-z\|_2 > r} \sqrt{p(z')}\, \exp\left(-\frac{\|z' - z\|_2^2}{8t}\right) \\
&\le \frac{1}{2\sqrt{p(z)}} e^{-r^2/(8t)} \sum_{z' \in \mathcal{C}} \sqrt{p(z')}.
\end{aligned}
$$

Define

$$S := \sum_{z \in \mathcal{C}} \sqrt{p(z)}.$$

By definition of $H_{1/2}$ we have $S^2 = e^{H_{1/2}}$. Averaging over $Z \sim p(\cdot)$,

$$
\begin{aligned}
\mathbb{P}(R > r) &= \sum_{z \in \mathcal{C}} p(z)\, \mathbb{P}(R > r \mid Z = z) \\
&\le \frac{S}{2} e^{-r^2/(8t)} \sum_{z \in \mathcal{C}} \sqrt{p(z)} \\
&= \frac{S^2}{2} e^{-r^2/(8t)} \\
&= \frac{1}{2} \exp\left(H_{1/2} - \frac{r^2}{8t}\right).
\end{aligned}
\tag{35}
$$

Let

$$r_0 := \sqrt{Ct\, H_{1/2}},$$

where $C \ge 0$ is a constant to be chosen later. Decompose

$$\mathbb{E}[R^4] = \mathbb{E}\left[R^4 \mathbf{1}\{R \le r_0\}\right] + \mathbb{E}\left[R^4 \mathbf{1}\{R > r_0\}\right].$$

On the set $\{R \le r_0\}$ we simply use $R^4 \le r_0^4$, hence

$$\mathbb{E}\left[R^4 \mathbf{1}\{R \le r_0\}\right] \le r_0^4 = C^2 t^2 H_{1/2}^2.
\tag{36}$$

For the tail part, we use (35):

$$\mathbb{E}\big[R^4 \mathbf{1}\{R > r_0\}\big] = \int_{r_0}^{\infty} 4r^3 \mathbb{P}(R > r)\,dr + r_0^4 \mathbb{P}(R > r_0)$$

$$\leq 2e^{H_{1/2}} \int_{r_0}^{\infty} r^3 e^{-r^2/(8t)}\,dr + \frac{C^2 t^2 H_{1/2}^2}{2} \exp\left(H_{1/2} - \frac{r_0^2}{8t}\right).$$

Evaluating the integral and plugging in $r_0 = \sqrt{Ct\,H_{1/2}}$, we get

$$\mathbb{E}\big[R^4 \mathbf{1}\{R > r_0\}\big] \leq \left[64t^2\Big(\frac{C}{8}H_{1/2} + 1\Big) + \frac{C^2 t^2 H_{1/2}^2}{2}\right] \exp\left(\Big(1 - \frac{C}{8}\Big)H_{1/2}\right). \tag{37}$$

Combining (36) and (37) yields

$$\mathbb{E}[R^4] \leq C^2 t^2 H_{1/2}^2 + \left[64t^2\left(\frac{C}{8}H_{1/2} + 1\right) + \frac{C^2 t^2 H_{1/2}^2}{2}\right] \exp\left(\Big(1 - \frac{C}{8}\Big)H_{1/2}\right).$$

This holds for all $C \geq 0$. In particular, for $C = 8$, we have

$$\mathbb{E}[R^4] \leq 96t^2 H_{1/2}^2 + 64t^2\left(H_{1/2} + 1\right).$$

This holds for all $t \geq 0$. $\qquad\square$

Combining Propositions C.1, C.3 and C.4, and using $\gamma = 1/t$, we obtain for every $\gamma > 0$,

$$|\mathrm{mmse}'(\gamma)| \leq \frac{96H_{1/2}^2 + 64\big(H_{1/2} + 1\big)}{\gamma^2}.$$

Moreover, when $H_{1/2} \geq c$, we have

$$H_{1/2} + 1 \leq \left(\frac{1}{c} + \frac{1}{c^2}\right) H_{1/2}^2,$$

and therefore

$$|\mathrm{mmse}'(\gamma)| \leq C_c \frac{H_{1/2}^2}{\gamma^2}, \qquad C_c := 96 + 64\left(\frac{1}{c} + \frac{1}{c^2}\right).$$

Setting $c = 2$ gives Corollary 4.9.

## D  Proof of Corollary 4.14

We bound Rényi entropy with Shannon entropy up to a constant, under a sub-exponential assumption on information content.

**Proposition D.1.** *Under Assumption 4.2 and 4.12,*

$$H_{1/2} \leq H + \frac{\nu^2}{2}. \tag{38}$$

*Proof.* Recall

$$S := \sum_{z \in \mathcal{C}} \sqrt{p(z)} = \mathbb{E}\big[e^{\imath(Z)/2}\big] = e^{H/2}\,\mathbb{E}\big[e^{(\imath(Z)-H)/2}\big].$$

By (SE) with $\lambda = \frac{1}{2}$ (which is allowed because $b \leq 2$ implies $1/2 \leq 1/b$),

$$\mathbb{E}\big[e^{(\imath(Z)-H)/2}\big] \leq e^{\nu^2/4},$$

and therefore
$$S \le e^{H/2} e^{\nu^2/4}.$$

Taking logarithms and recalling $H_{1/2} = 2 \log S$ yields

$$H_{1/2} = 2 \log S \le 2 \Big( \frac{H}{2} + \frac{\nu^2}{4} \Big) = H + \frac{\nu^2}{2},$$

which is (38).

$\square$

Combining with (9) and (10) leads to Corollary 4.14.

## E    Proof of Proposition 4.15

*Proof.* On $(s_{k-1}, s_k]$, the term $\big\| m_{T-s_{k-1}}(Y_{s_{k-1}}) - \hat{m}_{T-s_{k-1}}(Y_{s_{k-1}}) \big\|_2^2$ is $\mathcal{F}_{s_{k-1}}$-measurable, and

$$\int_{s_{k-1}}^{s_k} \frac{ds}{(T-s)^2} = \Big[ \frac{1}{T-s} \Big]_{s_{k-1}}^{s_k} = \gamma_k - \gamma_{k-1}.$$

Using $Y_{s_{k-1}} = X_{T-s_{k-1}} = X_{1/\gamma_{k-1}}$ and

$$m_{1/\gamma}(x) - \hat{m}_{1/\gamma}(x) = \frac{1}{\sqrt{\gamma}} \big( \hat{\varepsilon}_\gamma(x) - \varepsilon_\gamma^\star(x) \big),$$

we obtain

$$\mathcal{E}_{\mathrm{apx}} = \sum_{k=1}^{K} (\gamma_k - \gamma_{k-1}) \cdot \mathbb{E} \Big[ \frac{1}{\gamma_{k-1}} \big\| \varepsilon_{\gamma_{k-1}}^\star (X_{1/\gamma_{k-1}}) - \hat{\varepsilon}_{\gamma_{k-1}} (X_{1/\gamma_{k-1}}) \big\|_2^2 \Big],$$

which implies (14) under the SNR grid (12). $\square$

## F    Proof of Proposition 5.1

*Proof.* By Proposition 4.6,

$$\mathcal{E}_{\mathrm{disc}} = \sum_{k=1}^{K} \int_{\gamma_{k-1}}^{\gamma_k} \big( \mathrm{mmse}(\gamma_{k-1}) - \mathrm{mmse}(\gamma) \big) \, d\gamma$$

$$= \sum_{k=1}^{K} (\gamma_k - \gamma_{k-1}) \, \mathrm{mmse}(\gamma_{k-1}) \; - \int_{\gamma_0}^{\gamma_K} \mathrm{mmse}(\gamma) \, d\gamma. \tag{39}$$

Next, from the definition of $\mathcal{E}_{\mathrm{apx}}$, the integrand does not depend on $s$ except through $(T-s)^{-2}$, hence

$$\mathcal{E}_{\mathrm{apx}} = \sum_{k=1}^{K} \mathbb{E} \Big[ \big\| m_{T-s_{k-1}}(Y_{s_{k-1}}) - \hat{m}_{T-s_{k-1}}(Y_{s_{k-1}}) \big\|_2^2 \Big] \int_{s_{k-1}}^{s_k} \frac{ds}{(T-s)^2}. \tag{40}$$

But

$$\int_{s_{k-1}}^{s_k} \frac{ds}{(T-s)^2} = \Big[ \frac{1}{T-s} \Big]_{s_{k-1}}^{s_k} = \gamma_k - \gamma_{k-1}.$$

Also $Y_{s_{k-1}} = X_{T-s_{k-1}} = X_{1/\gamma_{k-1}}$, so with $t_{k-1} := T - s_{k-1} = 1/\gamma_{k-1}$,

$$\mathbb{E} \Big[ \big\| m_{T-s_{k-1}}(Y_{s_{k-1}}) - \hat{m}_{T-s_{k-1}}(Y_{s_{k-1}}) \big\|_2^2 \Big] = \mathbb{E} \Big[ \big\| m_{t_{k-1}}(X_{t_{k-1}}) - \hat{m}_{t_{k-1}}(X_{t_{k-1}}) \big\|_2^2 \Big].$$

Denote $m := m_{t_{k-1}}(X_{t_{k-1}}) = \mathbb{E}[Z \mid X_{t_{k-1}}]$, $\hat{m} := \hat{m}_{t_{k-1}}(X_{t_{k-1}})$. Then

$$\mathbb{E}\|Z - \hat{m}\|^2 = \mathbb{E}\|Z - m + m - \hat{m}\|^2$$
$$= \mathbb{E}\|Z - m\|^2 + \mathbb{E}\|m - \hat{m}\|^2 + 2\,\mathbb{E}\big[(Z - m)^\top (m - \hat{m})\big].$$

The cross term is zero by conditional orthogonality. Therefore,

$$\mathbb{E}\|m - \hat{m}\|^2 = \mathbb{E}\|Z - \hat{m}\|^2 - \mathbb{E}\|Z - m\|^2 = L_{k-1} - \mathrm{mmse}(\gamma_{k-1}).$$

Plugging into (40) yields

$$\mathcal{E}_{\mathrm{apx}} = \sum_{k=1}^{K} (\gamma_k - \gamma_{k-1})\Big(\mathcal{L}_{x_0}(\gamma_{k-1}) - \mathrm{mmse}(\gamma_{k-1})\Big). \tag{41}$$

Finally, add (39) and (41): the terms $\sum_{k=1}^{K}(\gamma_k - \gamma_{k-1})\,\mathrm{mmse}(\gamma_{k-1})$ cancel, giving (18). $\qquad\square$

## G    Schedule Optimization Algorithms

This appendix describes the algorithms used to compute the discretization schedule from a finite candidate set of SNR values. Let $\{\gamma_i\}_{i=0}^{n-1}$ be candidate SNRs (sorted increasing), and let $L(i)$ denote the estimated diagnostic risk at $\gamma_i$ (e.g., an $x_0$-prediction risk or a known rescaling of the $\varepsilon$-loss). We fix endpoints $i_0 = 0$ and $i_K = n - 1$. Define the regularized SNR axis

$$\eta(\gamma) := \frac{\gamma}{1 + \lambda^2\gamma}, \qquad \eta_i := \eta(\gamma_i), \qquad \ell_i := \log \gamma_i.$$

For a schedule $i_0 < i_1 < \cdots < i_K$, define log-SNR step sizes

$$h_k := \log \frac{\gamma_{i_k}}{\gamma_{i_{k-1}}} = \ell_{i_k} - \ell_{i_{k-1}}.$$

We minimize the surrogate objective

$$\sum_{k=1}^{K} (\eta_{i_k} - \eta_{i_{k-1}})\, L(i_{k-1}) \;+\; \alpha \sum_{k=2}^{K} (h_k - h_{k-1})^2, \tag{42}$$

where $\alpha = 0$ yields a first-order objective (e.g., DDIM), and $\alpha > 0$ adds the smoothness penalty used for second-order samplers (e.g., SDE-DPM++(2M)).

### G.1    Exact Dynamic Programming for $\alpha = 0$

When $\alpha = 0$, the objective in (42) becomes first-order (the cost of a step depends only on the current grid point) and can be solved exactly by a shortest-path dynamic program on a Directed Acyclic Graph (DAG). Algorithm 1 gives an $O(Kn^2)$ procedure that selects $K+1$ increasing indices $0 = i_0 < i_1 < \cdots < i_K = n-1$ by minimizing the transition costs $(\eta_{i_k} - \eta_{i_{k-1}})\, L(i_{k-1})$ with fixed endpoints.

### G.2    Heuristic Beam-and-Window Dynamical Programming (DP) for $\alpha > 0$

For $\alpha > 0$, the smoothness penalty couples consecutive log-steps: $(h_k - h_{k-1})^2$ depends on the triple $(i_{k-2}, i_{k-1}, i_k)$. An exact second-order DP over index pairs is possible but naively costs $O(Kn^3)$ due to the additional minimization over the predecessor at each transition. In practice, we use a fast approximate shortest-path routine that combines beam pruning with localized candidate expansion on the log-SNR axis. Algorithm 2 summarizes the resulting beam-and-window DP, which (i) maintains a bounded number of partial paths per endpoint (beam width $B$), and (ii) expands only candidate next indices within a window around a predicted next log-SNR location, optionally augmented with a small set of global candidates.

**Prediction rule** Given the last two indices $(a, b)$, we predict the next log-SNR via constant log-ratio continuation:

$$\ell_{\text{pred}} := \ell_b + (\ell_b - \ell_a) = 2\ell_b - \ell_a. \tag{43}$$

Algorithm 2 then expands a window of indices around the insertion position of $\ell_{\text{pred}}$ (with radius $W$), which targets schedules with approximately stable log-SNR ratios while keeping computation inexpensive.

## H    Experiments

This appendix collects supplementary material for the experimental section. We first record a simple identity that connects the diagnostic used in our schedule objective to the standard training loss in DDPM parameterizations. We then provide additional details for the toy GMM and ImageNet experiments reported in the main text.

*Remark* H.1. In the DDPM forward process

$$X_t = \sqrt{\bar{\alpha}_t}\, X_0 + \sqrt{1 - \bar{\alpha}_t}\, \varepsilon, \qquad \varepsilon \sim \mathcal{N}(0, I),$$

an $\varepsilon$-predictor $\hat{\varepsilon}_t(X_t)$ induces the usual $x_0$-predictor

$$\hat{X}_0(X_t) = \frac{X_t - \sqrt{1 - \bar{\alpha}_t}\, \hat{\varepsilon}_t(X_t)}{\sqrt{\bar{\alpha}_t}}.$$

Hence, with $\gamma_t := \frac{\bar{\alpha}_t}{1 - \bar{\alpha}_t}$ (SNR),

$$\|X_0 - \hat{X}_0(X_t)\|_2^2 = \frac{1 - \bar{\alpha}_t}{\bar{\alpha}_t}\, \|\varepsilon - \hat{\varepsilon}_t(X_t)\|_2^2 = \frac{1}{\gamma_t}\, \|\varepsilon - \hat{\varepsilon}_t(X_t)\|_2^2.$$

Therefore, the $x_0$-prediction risks appearing in our schedule objective can be obtained directly from the standard $\varepsilon$-training losses via a known SNR factor, with little extra post-training computation.

**Toy GMM experiments** We evaluate schedules on two 2D 8-component isotropic GMM priors with $\sigma_0 = 0.25$. `circle8` places means uniformly on a radius-4 circle with fixed non-uniform weights, and `grid8` places means on a $2 \times 4$ grid with fixed non-uniform weights. For each NFE $K \in \{5, 7, 10\}$, we compare the optimized schedule against linear-time and EDM ($\rho = 7$), using DDIM ($\eta = 1$) and SDE-DPM-Solver++(2M) samplers. We generate 20,000 samples per setting and report the negative mean log-likelihood under the true GMM (lower is better).

Tables 2–3 show LAS consistently achieves the best negative mean log-likelihood across both priors and both samplers, with the largest improvements at low NFE (notably $K = 5$ and $K = 7$). Linear-time is second-best, while EDM performs worst in these toy settings.

**ImageNet** $256 \times 256$ **(latent diffusion) details** We evaluate LAS on ImageNet $256 \times 256$ using a latent diffusion model with classifier guidance scale 2. We tune the SNR-axis regularization parameter by grid search over $\lambda \in \{0.5, 1.0, 1.5, 2.0\}$ using a pilot budget of 1,000 generated samples and fix the best value $\lambda = 1.5$ for all subsequent ImageNet experiments. For the DPM sampling, we additionally tune the exponent by grid search over $\alpha \in \{4, 8, 10, 12, 15\}$ (with all other settings fixed) and fix the best value $\alpha = 12$ for all DPM-Solver experiments.

For DDIM ($\eta = 1$), the LAS schedules (timesteps, noisy $\rightarrow$ clean) are:

$$K = 10: \ [999, 746, 607, 527, 462, 402, 342, 280, 208, 126, 0],$$

$$K = 20: \ [999, 808, 704, 635, 583, 543, 509, 479, 450, 422,$$
$$392, 362, 332, 300, 268, 234, 198, 158, 112, 60, 0].$$

Table 2: Negative mean log-likelihood (lower is better) on circle8.

| Sampler type | Sampler | Schedule | NFE=5 | NFE=7 | NFE=10 |
|---|---|---|---|---|---|
| Stochastic | DDIM ($\eta = 1$) | LAS | 1.937 | 1.639 | 1.609 |
| | | Linear-time | 4.063 | 2.408 | 1.810 |
| | | EDM | 6.748 | 4.013 | 2.546 |
| | SDE-DPM++(2M) | LAS | 1.721 | 1.500 | 1.574 |
| | | Linear-time | 3.314 | 2.085 | 1.662 |
| | | EDM | 5.585 | 7.272 | 3.897 |

Table 3: Negative mean log-likelihood (lower is better) on grid8.

| Sampler type | Sampler | Schedule | NFE=5 | NFE=7 | NFE=10 |
|---|---|---|---|---|---|
| Stochastic | DDIM ($\eta = 1$) | LAS | 2.077 | 1.758 | 1.650 |
| | | Linear-time | 4.310 | 2.516 | 1.849 |
| | | EDM | 5.832 | 4.047 | 2.604 |
| | SDE-DPM++(2M) | LAS | 1.875 | 1.569 | 1.586 |
| | | Linear-time | 3.553 | 2.188 | 1.689 |
| | | EDM | 5.319 | 6.713 | 3.820 |

For SDE-DPM-Sovler++(2M) and DPM-Solver++(2M), the LAS schedules are:

$$K = 10: \ [999, 848, 690, 553, 460, 380, 302, 210, 98, 14, 0],$$

$$K = 20: \ [999, 905, 804, 708, 627, 569, 523, 485, 448, 414,$$
$$380, 342, 306, 268, 226, 180, 126, 72, 26, 6, 0].$$

# I  Additional ablations for LAS

In this appendix we provide additional empirical evidence on the robustness of LAS and on the regime in which the KL-based schedule surrogate is most informative. All ablations in this section are computed using 5,000 generated samples. These experiments are intended to complement the main ImageNet results in Table 1, rather than to replace the full 50,000-sample evaluation used there.

## I.1  Qualitative ImageNet comparisons

Figures 1–3 provide qualitative ImageNet $256 \times 256$ comparisons in the low-NFE regime. For each comparison, we use the same pretrained model, guidance scale, class condition, number of function evaluations, and initial noise seed across schedules. These examples complement the quantitative FID, sFID, and IS results in Table 1.

## I.2  Comparison with Align Your Steps

We first compare LAS with Align Your Steps (AYS) (Sabour et al., 2024). The AYS numbers are taken from the corresponding paper under the same ImageNet $256 \times 256$ and NFE setting. Table 4 shows that LAS is competitive with AYS. In particular, for DDIM, LAS gives substantially better FID, sFID, and IS. For DPM-Solver++(2M), AYS gives slightly better FID, while LAS gives better sFID and IS. This comparison is useful because AYS is a strong schedule-optimization baseline, whereas LAS is obtained from training-loss information and does not require heavy post-training schedule optimization.

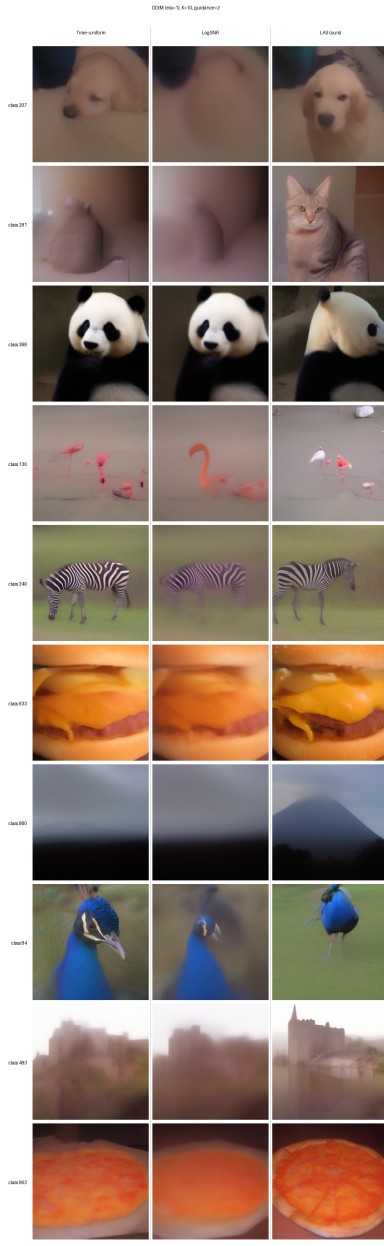

Figure 1: Qualitative ImageNet $256 \times 256$ comparison for DDIM ($\eta = 1$) at NFE= 10. We use matched class conditions and initial noise seeds across schedules.

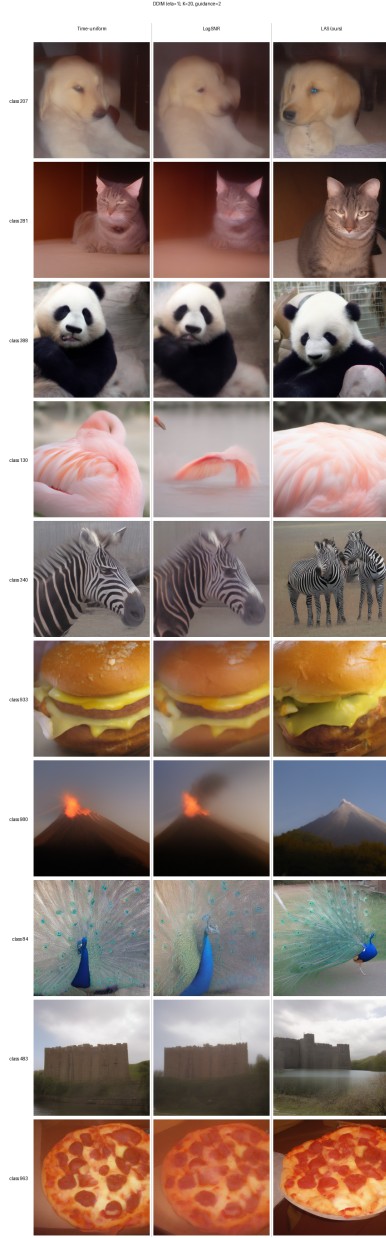

Figure 2: Qualitative ImageNet $256 \times 256$ comparison for DDIM ($\eta = 1$) at NFE= 20. We use matched class conditions and initial noise seeds across schedules.

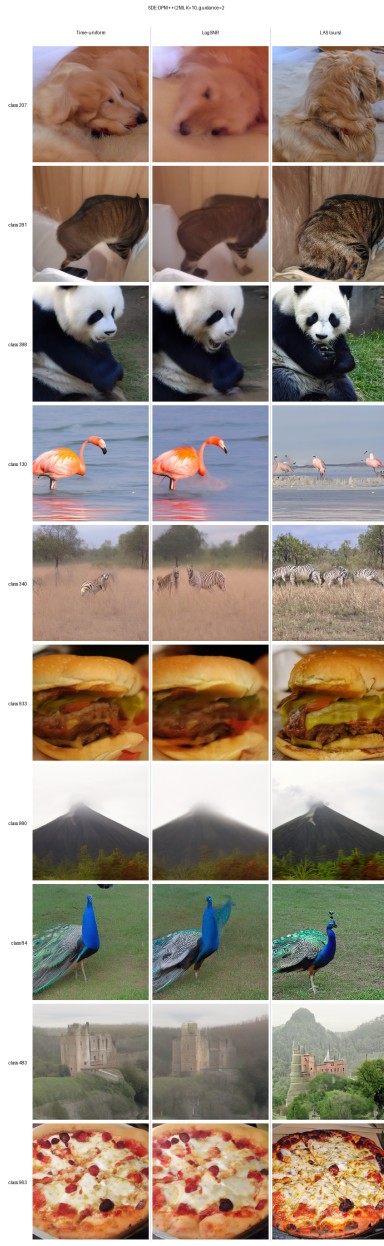

Figure 3: Qualitative ImageNet $256 \times 256$ comparison for SDE-DPM++(2M) at NFE= 10. We use matched class conditions and initial noise seeds across schedules.

---

**Algorithm 1** Exact schedule optimization for $\alpha = 0$ (first-order DP)

---

**Require:** Candidates $\{\gamma_i\}_{i=0}^{n-1}$ (increasing), risks $\{L(i)\}$, steps $K$, parameter $\lambda > 0$.
**Ensure:** Indices $0 = i_0 < i_1 < \cdots < i_K = n-1$ minimizing (42) with $\alpha = 0$.
1: Compute $\eta_i \leftarrow \gamma_i / (1 + \lambda^2 \gamma_i)$ for all $i$.
2: $\texttt{end} \leftarrow n-1$, INF large.
3: Allocate $\texttt{dp}[1{:}K, 0{:}\texttt{end}] \leftarrow$ INF and $\texttt{par}[1{:}K, 0{:}\texttt{end}] \leftarrow -1$. $\{\texttt{dp}[k,j]$: best cost to reach $j$ in exactly $k$ transitions from $0\}$
4: **for** $j = 1$ **to** $\texttt{end} - (K-1)$ **do**
5: $\quad \texttt{dp}[1,j] \leftarrow (\eta_j - \eta_0)\, L(0); \quad \texttt{par}[1,j] \leftarrow 0$.
6: **end for**
7: **for** $k = 2$ **to** $K-1$ **do**
8: $\quad \texttt{maxJ} \leftarrow \texttt{end} - (K-k)$
9: $\quad$ **for** $j = k$ **to** $\texttt{maxJ}$ **do**
10: $\quad\quad \texttt{dp}[k,j] \leftarrow \min\limits_{i<j} \left\{ \texttt{dp}[k-1,i] + (\eta_j - \eta_i)\, L(i) \right\}$.
11: $\quad\quad \texttt{par}[k,j] \leftarrow \arg\min\limits_{i<j} \left\{ \texttt{dp}[k-1,i] + (\eta_j - \eta_i)\, L(i) \right\}$.
12: $\quad$ **end for**
13: **end for**
14: $i_K \leftarrow \texttt{end}$.
15: $i_{K-1} \leftarrow \arg\min\limits_{i<i_K} \left\{ \texttt{dp}[K-1,i] + (\eta_{i_K} - \eta_i)\, L(i) \right\}$.
16: Backtrack $i_{K-2}, \ldots, i_0$ using $\texttt{par}$ and return $(i_0, \ldots, i_K)$.

---

Table 4: Comparison with Align Your Steps (AYS) on ImageNet $256 \times 256$. The AYS numbers are taken from Sabour et al. (2024).

| Sampler | Schedule | FID ↓ | sFID ↓ | IS ↑ |
|---|---|---|---|---|
| DDIM | LAS | **15.71** | **47.79** | **168.94** |
| DDIM | AYS | 23.13 | 64.37 | 118.61 |
| DPM-Solver++ (2M) | LAS | 4.59 | **5.74** | **263.69** |
| DPM-Solver++ (2M) | AYS | **4.31** | 6.64 | 260.32 |

## I.3 Sensitivity to the SNR-axis regularization

Recall that LAS optimizes the schedule on the regularized SNR axis

$$\gamma_{\text{reg}}(\gamma) = \frac{\gamma}{1 + \lambda^2 \gamma}.$$

The parameter $\lambda$ controls how strongly the high-SNR region is compressed. Table 5 reports the sensitivity of the resulting schedule to $\lambda$. The results show that LAS is not tied to a single finely tuned value: a range of values improves substantially over the time-uniform schedule. Very small values of $\lambda$ can overweight the high-SNR region, where the pathwise KL bound is less aligned with terminal sample quality, while moderate values give better performance.

## I.4 Sensitivity to the smoothness regularization

For second-order solvers, highly inhomogeneous step sizes can lead to instability. We therefore add the smoothness penalty

$$\alpha \sum_{k=2}^{K} (h_k - h_{k-1})^2, \qquad h_k := \log(\gamma_k / \gamma_{k-1}),$$

---

**Algorithm 2** Heuristic schedule optimization for $\alpha > 0$ (beam-pruned windowed DP)

---

**Require:** Candidates $\{\gamma_i\}_{i=0}^{n-1}$ (increasing), risks $\{L(i)\}$, steps $K$, $\lambda > 0$, $\alpha > 0$.
**Require:** Beam width $B$, window radius $W$, extra candidates $E$.
**Ensure:** Approximate minimizer of (42): indices $0 = i_0 < i_1 < \cdots < i_K = n - 1$.
 1: Compute $\eta_i \leftarrow \gamma_i/(1 + \lambda^2 \gamma_i)$ and $\ell_i \leftarrow \log \gamma_i$ for all $i$.
 2: $\texttt{end} \leftarrow n - 1$.
 3: A *state* is $(a, b, cost)$ representing the last two indices $(a, b)$ and accumulated cost.
 4: Let $\mathcal{S}_k(b)$ be a list of up to $B$ states that end at index $b$ after $k$ transitions.
 5: **for** $b = 1$ **to** $\texttt{end} - (K-1)$ **do**
 6:     $\mathcal{S}_1(b) \leftarrow \{(0, b, (\eta_b - \eta_0)L(0))\}$.
 7: **end for**
 8: **for** $k = 2$ **to** $K - 1$ **do**
 9:     $\texttt{maxIdx} \leftarrow \texttt{end} - (K - k)$.
10:     Initialize all $\mathcal{S}_k(\cdot)$ to empty.
11:     **for** each endpoint $b$ with $\mathcal{S}_{k-1}(b) \neq \emptyset$ **do**
12:         **for** each $(a, b, cost) \in \mathcal{S}_{k-1}(b)$ **do**
13:             $\ell_{\text{pred}} \leftarrow 2\ell_b - \ell_a$.
14:             $j \leftarrow \min\{j : \ell_j \geq \ell_{\text{pred}}\}$
15:             $\mathcal{C} \leftarrow \{c : \max(b+1, j-W) \leq c \leq \min(\texttt{maxIdx}, j+W)\}$.
16:             **for** each $c \in \mathcal{C}$ **do**
17:                 $\Delta_{\text{base}} \leftarrow (\eta_c - \eta_b) L(b)$.
18:                 $\Delta_{\text{sm}} \leftarrow \alpha\Big((\ell_c - \ell_b) - (\ell_b - \ell_a)\Big)^2$.
19:                 $\texttt{newCost} \leftarrow cost + \Delta_{\text{base}} + \Delta_{\text{sm}}$.
20:                 Insert $(b, c, \texttt{newCost})$ into $\mathcal{S}_k(c)$ with a backpointer to $(a, b)$.
21:             **end for**
22:         **end for**
23:     **end for**
24:     **Beam pruning:** for each $c$, keep only the $B$ states in $\mathcal{S}_k(c)$ with smallest cost.
25: **end for**
26: $i_K \leftarrow \texttt{end}$.
27: Among all states $(a, b, cost) \in \mathcal{S}_{K-1}(b)$, select the one minimizing

$$cost + (\eta_{i_K} - \eta_b)L(b) + \alpha\Big((\ell_{i_K} - \ell_b) - (\ell_b - \ell_a)\Big)^2.$$

28: Backtrack pointers to recover $(i_0, \ldots, i_K)$ and return.

---

Table 5: Sensitivity of LAS to the regularization parameter $\lambda$ using 5,000 samples.

| Method | FID $\downarrow$ |
|---|---|
| Time-uniform | 31.68 |
| LAS, $\lambda = 0.5$ | 42.91 |
| LAS, $\lambda = 1.0$ | 25.69 |
| LAS, $\lambda = 1.5$ | 22.23 |
| LAS, $\lambda = 2.0$ | **21.38** |
| LAS, $\lambda = 2.5$ | 21.95 |

as described in Section 6. Table 6 reports the effect of the smoothness weight $\alpha$. The results show that moderate smoothing improves performance relative to time-uniform sampling, while the performance is fairly stable over a range of $\alpha$ values.

Table 6: Sensitivity of LAS to the smoothness parameter $\alpha$ using 5,000 samples.

| Method | FID $\downarrow$ |
|---|---|
| Time-uniform | 14.23 |
| LAS, $\alpha = 0$ | 13.89 |
| LAS, $\alpha = 4$ | 12.83 |
| LAS, $\alpha = 8$ | 12.61 |
| LAS, $\alpha = 10$ | **12.40** |
| LAS, $\alpha = 12$ | 12.61 |
| LAS, $\alpha = 15$ | 12.48 |

Table 7: Mixed-optimization ablation using 5,000 samples. We optimize only the first $m$ reverse steps and keep the remaining steps time-uniform.

| Method | FID $\downarrow$ |
|---|---|
| LAS, $\lambda = 1.5$ | 22.23 |
| Time-uniform | 31.68 |
| Mixed, $m = 2$ | 31.44 |
| Mixed, $m = 4$ | 28.53 |
| Mixed, $m = 6$ | 23.44 |
| Mixed, $m = 8$ | 27.69 |
| Mixed, $m = 10$ | 238.95 |

## I.5 Where the schedule surrogate is informative

We next study where the KL-based schedule surrogate is most useful. In the mixed-optimization experiment, we optimize only the first $m$ reverse steps using LAS and keep the remaining steps time-uniform. Thus, small $m$ values modify only the early part of the reverse trajectory, while larger $m$ values push the optimization further toward the high-SNR terminal region.

Table 7 shows a non-monotone trend. Optimizing the early reverse steps improves over the time-uniform baseline, suggesting that the surrogate is informative in the low-SNR part of the trajectory. However, pushing the optimization too far toward the terminal high-SNR regime can hurt performance substantially. This supports the interpretation that the pathwise KL upper bound is most useful for schedule design away from the near-terminal regime, where the data-processing step from pathwise KL to terminal sample quality can be loose.

Overall, these ablations support three conclusions. First, LAS is competitive with AYS while using substantially lighter post-training computation. Second, the regularization parameters are not extremely sensitive, since a range of moderate values improves over the corresponding heuristic schedules. Third, the KL-based surrogate is most informative in the low- to intermediate-SNR part of the trajectory, while near-terminal high-SNR optimization requires regularization or smoothing.

