# OpenReview forum: "Entropy-Controlled Time-Discretization Bounds and Loss-Adaptive Schedules for Diffusion Models"
_TMLR — Under review for TMLR_

### Review · Reviewer_ewhy · 2026-06-20

**Summary Of Contributions:**

This paper proposes an information-theoretic approach in the convergence analysis of discretized diffusion model, which is not only ambient dimension free but also free of geometric assumptions on the target distribution. More specifically, the authors break down the KL divergence sampling error into three distinct components: initialization error, score estimation error, and time discretization error. They then formulate the discretization error exactly as a minimum mean-square error (MMSE) functional. They then establish a discretization error bound, which is free of  ambient dimension d, in the price of introducing a different assumption:  finite order-1/2 Rényi entropy.

Empirically, the authors propose a Loss-Adaptive Schedule (LAS) for discretizing the reverse SDE whose effect is supported by a single  ImageNet 256 × 256 dataset with Latent Diffusion.

**Strengthness**

1. The paper is well-written and easy to follow. Related works are appropriately cited.
2. The experiments do support a practical positive effect of LAS, especially for coarse, low-NFE sampling

**Weakness**
1. My main concern is the writing of proposition 3.1. More specifically, (a) what is $P_{0}$ and $\tilde{P}_{0}$ in the bottom of page 4?  (b) could the authors elaborate why the identity (3) in exact?
2. The baselines are not well tuned, leading to the possible unfair compariosn. LAS uses training-loss information, but $\lambda$ and $\alpha are tuned using generated samples/FID. The budget is small, but it is still sample-quality tuning. The baselines do not appear equally tuned.

**Additional Comments:**

Can LAS facilitate the sample generation speed? Like the one-step diffusion model.

**Audience:**

Yes

**Audience Explanation:**

The effective discretizing schedule of diffusion model proposed by this paper could lead to better sample quality of diffusion and speed up the sample generation.

Readers with theoretical background may find the conclusion that the convergence of discretizing diffusion model can be free of ambient dimension d in the price of finite Renyi entropy interesting since in provides another view of convergence analysis.

**Broader Impact Concerns:**

I have no broader impact concern of this worl, since most parts are theoreritical results.

**Claims And Evidence:**

Yes

**Claims Explanation:**

See weakness 1 and 2. My major concern is the plausible writing of proposition 3.1. The authors are encouraged to answer my questions in weakness 1.

Edit: My major concern is resolved by authors response.

**Requested Changes:**

1. Rewrite the derivation and statements about proposition 3.1.
2. Double check the notations, make sure they are consistent.

---

> ### Author Response · Authors · 2026-06-20
> **Clarification of Proposition 3.1 and LAS Sampling**
>
> We thank the reviewer for the careful reading. We first emphasize that the issue in Proposition 3.1 is only a presentation/notation issue and does not affect any of our main results. The dimension-free result concerns the discretization term
> $\mathcal{E}\_{\mathrm{disc}}$, whose MMSE representation and Renyi-entropy bound remain unchanged. The LAS algorithm is also unchanged, since it is derived from the schedule-dependent part $\mathcal{E}\_{\mathrm{disc}}+\mathcal{E}\_{\mathrm{apx}}$.
>
> Regarding $\mathbb{P}\_0$ and $\widetilde{\mathbb{P}}\_0$, these denote the initial marginals of the exact and implemented reverse processes. In our notation, $\mathbb{P}\_0=p\_T=\mathcal{L}(X\_T)$, while the implemented sampler is initialized from $\widetilde{\mathbb{P}}\_0=\mathcal{N}(0,T I\_d)$. We will define these quantities explicitly and use consistent notation. Thank you for your comment!
>
> Regarding the exactness of Eq. (3), we agree that the current wording is misleading. The KL itself should be stated as an upper bound:
> $$
> \mathrm{KL}(\mathbb{P}\_{T\_\delta} \| \widetilde{\mathbb{P}}\_{T\_\delta})
> \le
> \mathrm{KL}(\mathbb{P}\_0 \| \widetilde{\mathbb{P}}\_0)
> +
> \frac{1}{2}
> \sum\_{k=1}^K
> \mathbb{E}\_{\mathbb{P}}\left[
> \int\_{s\_{k-1}}^{s\_k}
> \left\|
> \beta\_s(Y\_s)-\widetilde{\beta}\_s^{(k)}(Y\_s)
> \right\|^2 ds
> \right].
> $$
>
>
> The exactness we intended in Eq. (3) is the exact decomposition of the quadratic drift-mismatch term. After adding and subtracting $m\_{T-s\_{k-1}}(Y\_{s\_{k-1}})$, the drift mismatch splits into a discretization component and an approximation component:
> $$
> A\_s=\frac{m\_{T-s}(Y\_s)-m\_{T-s\_{k-1}}(Y\_{s\_{k-1}})}{T-s},
> \qquad
> B\_s=\frac{m\_{T-s\_{k-1}}(Y\_{s\_{k-1}})-\widehat{m}\_{T-s\_{k-1}}(Y\_{s\_{k-1}})}{T-s}.
> $$
> The cross term vanishes by orthogonality: $B\_s$ is measurable with respect to the previous gridpoint information, while
> $$
> \mathbb{E}\_{\mathbb{P}}\left[
> A\_s \mid \mathcal{F}\_{s\_{k-1}}
> \right]=0
> $$
> by the tower property for the posterior mean. Hence,
> $$
> \mathbb{E}\_{\mathbb{P}}\left[\left\|A\_s+B\_s\right\|^2\right]=\mathbb{E}\_{\mathbb{P}}\left[\left\|A\_s\right\|^2\right]+\mathbb{E}\_{\mathbb{P}}\left[\left\|B\_s\right\|^2\right].
> $$
> Therefore,
> $$
> \frac{1}{2}
> \sum\_{k=1}^K
> \mathbb{E}\_{\mathbb{P}}\left[\int\_{s\_{k-1}}^{s\_k}\left\|\beta\_s(Y\_s)-\widetilde{\beta}\_s^{(k)}(Y\_s)\right\|^2 ds\right]=\frac{1}{2}\left(\mathcal{E}\_{\mathrm{disc}}+\mathcal{E}\_{\mathrm{apx}}\right)
> $$
> exactly. We will rewrite Proposition 3.1 to make this distinction clear: the KL is upper bounded, while the quadratic drift-mismatch term decomposes exactly.
>
> Regarding the LAS comparison, the standard schedules we compare against, such as time-uniform, LogSNR, are fixed heuristic schedules. They do not provide a framework for improving the schedule using information from the trained model. In contrast, LAS provides such a framework: it uses the training-loss profile to adapt the allocation of sampling steps across noise levels. Thus, LAS is not simply another fixed baseline; it is a loss-adaptive schedule-selection method that creates room for improvement after training. Our point is other schedulers have no room for improvement, however ours has.
>
> We also compare with Align Your Steps (AYS) in Appendix I. AYS is a strong schedule-optimization method, but it requires substantially heavier post-training computation. LAS is competitive with AYS while using much lighter information, since it relies on the training-loss profile. Appendix I also includes ablations over $\lambda$ and $\alpha$, showing that LAS is stable over a reasonable range of these parameters. This supports that the improvement is not due to a fragile hyperparameter choice.
>
> Also, LAS is not a one-step diffusion or distillation method. Its role is to improve sample quality at a fixed number of function evaluations, especially in low-NFE regimes. Therefore, LAS can facilitate faster generation by reducing the number of sampling steps needed to reach a target quality, and it is complementary to one-step or distillation-based methods.

---

> > ### Comment · Reviewer_ewhy · 2026-07-12
> >
> > I thank the updates from authors. The new derivations of Proposition 3.1 makes sense to me now. I am also satisfied with the clarifications in Appendix I. Overall this is an interesting paper connects statistics with Diffusion model.

---

### Review · Reviewer_wkqX · 2026-06-26

**Summary Of Contributions:**

The paper has two main contributions. It starts by defining an upper bound on the KL divergence between the terminal distribution, i.e., the generated and real data distributions, as the integrated drift mismatch along the sampling trajectory. Then, the integrated drift error is decomposed into (1) initialization error, (2) discretization error under the true denoiser, and (3) approximation error of the denoiser.

- While previous research involves dimensionality in the bound for (2) the discretization error, which creates a gap from reality, the paper derives a new error bound involving the entropy of the target distribution. This provides a new insight into the convergence of diffusion models. The error bound also suggests that the total error should be treated separately for each sampling trajectory.
- The paper also derives upper bounds for (2) the discretization error and (3) the approximation error of the denoiser, which encompass the training loss of diffusion models. Based on this, the paper proposes a loss-adaptive sampling schedule that assigns dense timesteps to regions where the error is higher and sparse timesteps elsewhere.

The paper is technically sound and insightful. In particular, the authors connect information theory with the convergence analysis of diffusion models, making the interpretation of the error bound clearer and requiring fewer assumptions. All derivation results are provided, and the experimental results, including a toy example and a latent Stable Diffusion model on ImageNet 256×256, demonstrate that the proposed method is also practical.

The paper has a few weaknesses.

- First, the paper assumes the target distribution to be discrete and finite, which is not generally acceptable. Although the authors provide a following remark stating that the assumption holds for latent diffusion models utilizing Vector-Quantized (VQ) encoders and decoders, most current diffusion models, regardless of domain, leverage continuous spaces. Thus, the theoretical validity is limited.
- Second, while the proposed new discretization error bound is insightful, the connection between loss-adaptive scheduling and the entropy-based error bound seems to be somewhat independent. In particular, the motivation for trajectory-dependent adaptiveness comes from the approximation error term, which is not directly related to the proposed entropy-based error bound. While I believe both sections are sufficiently meaningful, this somewhat limits the coherence of the paper.
- Third, the paper does not demonstrate any qualitative results.

**Audience:**

Yes

**Audience Explanation:**

The paper would be of interest to at least some members of TMLR’s audience who work on diffusion theory, sampling algorithms, and efficient generation. The entropy-based discretization error bound provides a useful perspective beyond previous dimension-dependent analyses, and it connects diffusion convergence analysis with information-theoretic quantities. The proposed loss-adaptive sampling schedule may also be practically relevant for improving diffusion sampling efficiency, as it is compatible with existing fast samplers such as DPM-Solver.

**Broader Impact Concerns:**

I do not identify major broader impact concerns beyond those already discussed.

**Claims And Evidence:**

Yes

**Claims Explanation:**

The paper provides sufficient derivations for the proposed KL divergence upper bound, the drift-error decomposition, and the entropy-based discretization error bound. In particular, it first rewrites the drift error as a difference between Tweedie’s estimates, one at a discretized timestep and the other at a continuous timestep. Then, the paper connects this difference to the difference between MMSEs, and finally relates the result to the derivative of the MMSE, which is bounded by Rényi entropy. These derivations are mostly clear and support the main theoretical claims. The connection between the total error bound and the diffusion training loss is also established.

For the proposed loss-adaptive sampling schedule, the toy experiments report negative log-likelihood, providing quantitative evidence that the generated samples better match the target distribution. The LDM experiments are conducted with three widely used samplers and multiple NFEs (10 and 20), providing empirical support for the applicability of the method.

Overall, I think the main claims are supported by the provided theoretical and empirical evidence, although the scope of the theoretical claims and following experiment is somewhat limited by the discrete and finite target distribution case.

**Requested Changes:**

I consider the following points non-critical. They are not necessary to secure my recommendation for acceptance, but addressing them would strengthen the paper.

- First, while the paper already discusses the relevance of the discrete/countable assumption for latent diffusion models with Vector-Quantized (VQ) encoders and decoders, most current diffusion and flow-based generative models operate in continuous spaces. Thus, it would be helpful to further clarify whether this assumption is primarily a technical requirement for the proof, or whether extending the analysis to continuous target distributions would require fundamentally different techniques. Such discussion would help readers better understand the applicability and scope of the theoretical results.

- Second, while the entropy-based discretization error bound is insightful, the connection between the theoretical analysis in Section 4 and the practical scheduler design in Section 5 seems somewhat indirect. In particular, the information-theoretic error bound is not directly used in the design of the loss-adaptive sampling schedule. While I believe both parts are sufficiently meaningful, a brief discussion of how these two perspectives complement each other would improve the overall coherence of the paper.

- Third, could the authors provide additional discussion on the choice of the regularized SNR transformation? While the motivation for compressing the high-SNR regime is well explained, the particular functional form appears to be a practical design choice. It would be helpful to clarify whether it is mainly motivated by empirical observations or supported by additional theoretical or empirical considerations.

- Finally, could the authors provide qualitative results for the proposed loss-adaptive sampling schedule? Although the quantitative results are useful, representative generated samples or visual comparisons would help readers better understand the practical effect of the proposed schedule on sample quality.

---

> ### Author Response · Authors · 2026-07-19
>
> We thank the reviewer for the constructive suggestions. We address each point below.
>
> **1. Discrete/countable assumption.**
> The countable-support assumption is needed only for the entropy-based MMSE derivative bound and the resulting discretization theorem. It is not required for the KL decomposition, the exact MMSE identity, Proposition 5.1, or LAS. Our proof uses posterior probability masses and sums over the support, so extending this particular bound to general continuous distributions would require different techniques, such as quantization arguments or additional regularity assumptions involving continuous information quantities. We now clarify this scope in the manuscript.
>
> **2. Connection between the entropy analysis and LAS.**
> The two contributions arise from the same KL decomposition but address different terms. The entropy analysis controls the intrinsic discretization error under the true denoiser. In practice, however, the schedule-dependent error also includes the learned-denoiser approximation error. Proposition 5.1 expresses their combined contribution through the model’s prediction-loss profile, which motivates LAS. Thus, LAS is complementary to, rather than directly derived from, the Rényi-entropy bound. We added this explanation to Section 5.
>
> **3. Regularized SNR transformation.**
> The transformation
>
> $$
> \gamma_{\mathrm{reg}}(\gamma)=
> \frac{\gamma}{1+\lambda^2\gamma}
> $$
>
> is motivated by the effective SNR when the target itself contains additive Gaussian noise. If the clean target is $X$ and the observed target is
>
> $$
> Z=X+\lambda\xi,
> \qquad
> \xi\sim\mathcal{N}(0,I),
> $$
>
> then adding diffusion noise with variance $1/\gamma$ produces total noise variance
>
> $$
> \lambda^2+\frac{1}{\gamma}.
> $$
>
> The effective SNR relative to the clean target is therefore
>
> $$
> \gamma_{\mathrm{eff}}=\frac{\gamma}{1+\lambda^2\gamma}.
> $$
>
> This motivates the functional form: when there is an irreducible noise level, increasing the nominal SNR should have progressively less influence. Our goal is to smoothly reduce the impact of the high-SNR region rather than truncate it abruptly. The transformation is monotone and approximately linear at low SNR, while gradually saturating at high SNR. It therefore preserves the ordering of noise levels while preventing near-terminal points from dominating the scheduling objective. This remains a practical design choice rather than a consequence of the main theorem, and we added sensitivity experiments in Appendix I.
>
> **4. Qualitative results.**
> We added qualitative ImageNet $256\times256$ comparisons in Appendix I. The comparisons use matched class conditions, guidance scales, NFEs, and initial noise seeds across schedules, complementing the quantitative FID, sFID, and IS results.

---

> > ### Comment · Reviewer_wkqX · 2026-07-21
> > **Official comments by the reviewer wkqX**
> >
> > Thanks to the authors for their kind response. My major requests have been properly addressed.
> >
> > - I agree that the discrete assumption is required only for the MMSE part. Since the paper clearly states this, my concern has been addressed.
> >
> > - Thanks for the clarification on LAS. My point was that the two contributions, which are derived from the same KL decomposition, seemed to be disconnected, as one would expect the error bound to provide some theoretical support for LAS. I agree that LAS is complementary to the error bound.
> >
> > - Thanks for the clarification on the design of the regularized SNR transformation.
> >
> > - Appendix I provides informative results.
> >
> > Overall, I think this is a well-written paper.

---

### Review · Reviewer_2KB2 · 2026-07-06

**Summary Of Contributions:**

**Summary:**

This paper analyzes the time-discretization error of reverse-time diffusion samplers. It decomposes the pathwise KL error into initialization, denoiser approximation, and discretization terms. The main technique is to represent the time-discretization term as an area gap of the MMSE for the Gaussian channel induced by the Brownian forward process. The paper then bounds the derivative of this MMSE using the order-$1/2$ R\'{e}nyi entropy of the target distribution. The resulting bound is proved for target distributions with finite second moment, discrete support contained in a countable set, and finite $H_{1/2}$. Motivated by the same KL decomposition, the paper proposes a Loss-Adaptive Schedule (LAS), which selects the sampling grid using an empirical loss profile over noise levels, equivalently an $x_0$-prediction risk related to standard training losses. Numerical experiments are reported on toy Gaussian mixtures and ImageNet $256\times256$ latent diffusion models.

**Audience:**

Yes

**Audience Explanation:**

I believe some contributions have their own value, and at least some individuals in TMLR's audience will be interested. See overall evaluation.

**Claims And Evidence:**

Yes

**Claims Explanation:**

*To be more precise, some contributions are overclaimed.*

**Overall Evaluation:**

The paper proposes an interesting mathematical idea. In particular, Proposition 4.5 gives an exact MMSE representation of the discretization error, and Theorem 4.7 gives an entropy-based bound on the MMSE derivative under finite-moment and countable-support assumptions. However, the contributions are relatively overclaimed than what the existing theoretical and experimental results support. From my perspective, the main theoretical contribution should be described as an entropy-dependent KL discretization analysis for countable target distributions, together with a practical loss-based schedule. This is somehow more restrictive than a general dimension-free convergence result: as Remark 4.8 states, the bound removes explicit ambient dimension from the displayed discretization estimate, but replaces it by an entropy quantity that may still depend on the underlying dimension.

Overall, I think this submission is not ready for acceptance in its current form. Under the TMLR's standard, the main concern is whether its contributions are supported by clear and convincing theoretical and experimental evidence. In particular, the paper overclaims the scope of the dimension-free guarantee, does not fully specify how the discrete theory applies to the ImageNet latent setting, leaves the learned denoiser error as an uncontrolled term, and presents LAS more strongly than the proved theory supports. A revised version could be suitable if the authors properly narrow the contributions, discuss the entropy dependence and support assumptions more explicitly, explain precisely when the ImageNet latent setting satisfies the theoretical assumptions/results, and present LAS as a competitive scheduling method rather than as uniformly superior.

**Requested Changes:**

**Part 1.**

*The dimension-free claim is overstated.* Theorem 4.7 proves $|\mathrm{mmse}'(\gamma)| \leq C H_{1/2}^{2}/\gamma^{2}$ under Assumptions 4.1--4.2 and the condition $H_{1/2}>c>0$. This yields the discretization bound in Eq. (9), and same $H_{1/2}^{2}$ dependence appears in the discretization part of the final KL bound in Eq. (16). Remark 4.8 correctly states that the result is not distribution-free: the displayed bound has no explicit ambient dimension $d$, but the complexity is measured by an entropy quantity that may itself depend on $d$. Therefore, the result should not be stated as removing dimension dependence in general. For example, if $p$ is uniform on $\{0,1\}^{d}$, then $H_{1/2}=d\log 2$. Thus, even ignoring the initialization and denoiser-approximation terms, the discretization contribution in Eq. (16) scales as $O \left(\frac{d^{2}(\log\Lambda)^{2}}{K}\right)$, up to constants. The initialization term $\gamma_{0}M_{2}/2$ may also depend on $d$. The abstract and introduction should say ``entropy-dependent with no explicit ambient dimension'', not broadly dimension-independent. The paper should also report $H_{1/2}$, or upper bounds on it, for representative discrete distributions and for the code distribution used in the experiments.

**Part 2.**

*The connection between the discrete theorem and the ImageNet experiments is not fully specified.* The entropy-based MMSE derivative bound uses Assumption 4.2, which assumes support on a countable set $C\subset\mathbb{R}^d$. The proof of Proposition C.4 also uses sums over $C$, in particular in the tail estimate leading to Eq. (28). Remark 4.3 and Section 7 explain that VQ-based latent diffusion gives finite support after mapping codebook indices to Euclidean embeddings. This is a valid setting for Assumption 4.2. However, Section 6.2 and Appendix H do not give enough details to verify the theorem-level connection for the ImageNet $256\times256$ experiments. The authors should state the exact first-stage latent model used in the experiments, including whether it is VQ-regularized or KL-regularized, the codebook size, the number of latent positions, and the induced support of the full latent variable. They should also explain how $H_{1/2}$ is bounded or estimated for the resulting code distribution. If the evaluated latent representation is continuous, then Theorem~4.7 does not apply directly; a quantization or covering argument would be needed, with possible additional entropy or covering-number dependence.

**Part 3.**

*Control learned-denoiser error.* Proposition 4.13 defines per-level $\epsilon$-prediction errors $\epsilon_k$ and shows that, under the geometric SNR grid, $E_{\rm apx}$ is proportional to $\sum_{k=1}^K\epsilon_k$. The final KL estimate in Eq. (16) contains $\log\Lambda\cdot K^{-1}\sum_{k=1}^K\epsilon_k$. The paper does not provide a direct theoretical bound on these errors in terms of sample size, model class, optimization accuracy, or dimension. Hence, the final KL estimate depends on accurate learned denoiser. It is not an end-to-end convergence theorem for learned diffusion models. This distinction should be made clear in the title, abstract, and theorem discussion. Otherwise, the KL bound may be uninformative when the learned denoiser has large prediction error.

**Part 4.**

*LAS is only partly justified by the proved theory.* The paper first derives the geometric, or log-linear, SNR grid in Eq. (11) as the minimizer of the entropy-based upper bound for $E_{\rm disc}$. Section 5 then notes that optimizing $E_{\rm disc}$ alone is limited because the pathwise KL also contains $E_{\rm apx}$. Proposition 5.1 gives an exact identity for $E_{\rm disc}+E_{\rm apx}$ in terms of the $x_0$-prediction risk, which supports a loss-based objective on the original SNR axis. However, the implemented LAS then uses the regularized axis $\gamma_{\rm reg}(\gamma)=\gamma/(1+\lambda^2\gamma)$ in Eq. (18) and adds the smoothness penalty in Eq. (19) for second-order solvers. The paper motivates these choices by the empirical looseness of the pathwise KL near high SNR. These choices may help in practice, but they are not direct consequences of the entropy-based discretization theorem. The paper should separate the proved geometric-grid result, the exact loss-based identity, and the empirical regularization used in LAS.

**Part 5.**

*The assumptions and constants need clearer reporting.* Theorem 4.7 requires finite second moment, countable support, finite $H_{1/2}$, and $H_{1/2}>c>0$. The constant $C$ in $|\mathrm{mmse}'(\gamma)|\leq C H_{1/2}^2/\gamma^2$ is not made explicit, and the dependence on the lower threshold $c$ is not discussed. This matters since the proof of Proposition C.4 absorbs a lower-order term of order $t^2(H_{1/2}+1)$ into $t^2H_{1/2}^2$ using $H_{1/2}>c$. Corollary 4.12 replaces $H_{1/2}$ by Shannon entropy only under Assumption 4.10, which requires sub-exponential information content. These are not geometric assumptions, but they are still distributional assumptions. The paper should list these requirements when the main claim is made and should state whether constants depend on $c$.

---

> ### Author Response · Authors · 2026-07-19
>
> We thank the reviewer for the careful reading and constructive comments. The comments helped us sharpen the presentation and distinguish more explicitly among the time-discretization analysis, learned-denoiser error, and practical construction of LAS.
>
> To make the precise scope of the theoretical result immediately clear, we changed the title to:
>
> **Entropy-Controlled Time-Discretization Bounds and Loss-Adaptive Schedules for Diffusion Models.**
>
> We also revised the abstract, introduction, contribution summary, theorem discussion, experimental section, and limitations.
>
> ### 1. Scope of the dimension-free claim
>
> Our use of “dimension-free” follows a common convention in diffusion-sampling theory: the absence of explicit ambient-dimension dependence in the sampling-complexity or numerical-discretization analysis of the reverse dynamics, often conditional on exact or sufficiently accurate score access. It does not mean that every component of learning and sampling is dimension-independent. Recent dimension-free analyses similarly focus on numerical simulation or reverse-process discretization while treating score-approximation and other errors separately.
>
> Nevertheless, to remove any ambiguity, we now use the more precise phrase **“entropy-controlled time-discretization bounds with no explicit ambient-dimension dependence.”** Theorem 4.7 controls the time-discretization component of the KL decomposition. Initialization and learned-denoiser errors remain separate, explicit terms, and the result is not presented as an end-to-end statistical-learning or sample-complexity theorem.
>
> Replacing direct ambient-dimension dependence by $H_{1/2}$ is meaningful because entropy is a property of the probability law, not of the Euclidean coordinates used to represent its outcomes. In particular, an injective re-embedding of the same discrete distribution into a higher-dimensional space leaves its Rényi entropy unchanged.
>
> We recognize that $H_{1/2}$ may itself scale with support size, sequence length, or other problem parameters. For $p$ supported on a finite set $\mathcal{C}$, with $N=|\mathcal{C}|$, define the order-$1/2$ Rényi entropy in bits by
>
> $$H_{1/2}^{(2)}(p)=2\log_2\Bigg(\sum_{z\in\mathcal{C}}\sqrt{p(z)}\Bigg).$$
>
> If $u$ is uniform on $\mathcal{C}$, then
>
> $$H_{1/2}^{(2)}(p)=\log_2 N+2\log_2 A(p,u),$$
>
> where $A(p,u)=\sum_{z\in\mathcal{C}}\sqrt{p(z)u(z)}$ is the Bhattacharyya affinity. Equivalently,
>
> $$H_{1/2}^{(2)}(p)=\log_2 N+2\log_2\Big(1-H_{\mathrm{Hel}}^2(p,u)\Big)\leq\log_2 N,$$
>
> with equality if and only if $p=u$. This parallels
>
> $$H^{(2)}(p)=\log_2 N-\operatorname{KL}_2(p|u)\leq\log_2 N.$$
>
> Thus, the reviewer’s uniform example is the worst case for a fixed support. For a nonuniform distribution, $H_{1/2}$ is strictly smaller than the logarithm of the support size.
>
> ### 2. Connection to the ImageNet experiments and the entropy scale
>
> We added the exact first-stage specification for the ImageNet experiment. We use the class-conditional ImageNet LDM-VQ-8 model, whose first stage is VQ-regularized with downsampling factor $f=8$. A $256\times256$ image is represented by a $32\times32$ grid, giving $L=32\cdot32=1024$ discrete latent positions. The codebook has $S=16384=2^{14}$ entries, with four-dimensional embeddings. Thus, if $J=(J_1,\ldots,J_{1024})\in[16384]^{1024}$ denotes the code sequence and $Z=\Phi(J)$ its flattened embedding, then $Z\in\mathbb{R}^{4096}$.
>
> Although $Z$ lies in a 4,096-dimensional Euclidean space, it has finite support and therefore satisfies Assumption 4.2. The official configuration specifies a VQ first stage with 16,384 codebook entries, embedding dimension four, and $256\times256$ input resolution.
>
> The immediate support bound is $|\operatorname{supp}(Z)|\leq(2^{14})^{1024}=2^{14336}$, and therefore $H_{1/2}^{(2)}(Z)\leq14336$ bits.
>
> This verifies theorem applicability, but it is a very loose entropy estimate because it treats every possible VQ sequence as attainable and ignores the strong dependence and nonuniformity among latent positions.
>
> A more informative way to assess the entropy scale of the ImageNet distribution is to examine the most compact high-fidelity discrete representation available. Entropy is invariant under an injective representation: if $C=E(X)$ is an injective encoding of an image-valued random variable $X$, then $H_{1/2}(X)=H_{1/2}(C)$.
>
> More generally, if the representation is approximately information-preserving, the entropy capacity of its code provides a practical upper proxy for the effective entropy of the image distribution at that reconstruction fidelity.
>
> This motivates our use of TiTok, a compact discrete representation for ImageNet reconstruction and generation that represents a $256\times256$ image with only 32 discrete tokens. TiTok-L-32 uses a codebook of size $4096=2^{12}$, so its full representation has at most $(2^{12})^{32}=2^{384}$ possible values. Consequently, $H_{1/2}^{(2)}(Z_{\mathrm{TiTok}})\leq384$ bits.

---

> ### Author Response · Authors · 2026-07-19
>
> ### 2. Connection to the ImageNet experiments and the entropy scale (continued)
>
> A deterministic decoder cannot increase order-$1/2$ Rényi entropy. Thus, 384 bits rigorously upper-bounds the TiTok-reconstructed ImageNet distribution. Because TiTok retains strong ImageNet reconstruction and generation quality with this compact code, we use this quantity as an empirically grounded upper proxy for the effective entropy of the ImageNet distribution at TiTok’s reconstruction fidelity.
>
> TiTok therefore provides a meaningful upper bound for a high-quality approximation of the ImageNet distribution and a substantially sharper indication of its effective information scale than the naive VQ support bound.
>
> This is the theorem’s central interpretation. The pixel representation has ambient dimension $256\cdot256\cdot3=196608$; the experimental VQ representation gives a formal but loose upper bound of 14,336 bits; and a compact high-fidelity representation gives an upper proxy of 384 bits. Entropy dependence can reflect representation efficiency, whereas direct ambient-dimension dependence cannot.
>
> We also clarify that countable support and finite $H_{1/2}$ are required only for the entropy-based MMSE derivative bound. They are not required for the KL decomposition, exact MMSE representation, or LAS, subject to the corresponding moment and integrability conditions.
>
> ### 3. Learned-denoiser error
>
> The paper does not claim an end-to-end statistical-learning guarantee, and we do not bound the learned-denoiser error in terms of sample size, model class, optimization accuracy, or dimension.
>
> The KL decomposition separates $E_{\mathrm{init}}$, $E_{\mathrm{apx}}$, and $E_{\mathrm{disc}}$, corresponding to initialization, learned-denoiser approximation, and time-discretization error. The per-level prediction errors $\epsilon_k$ remain explicit in the final KL estimate and are not absorbed into the discretization analysis.
>
> We now state throughout that the theoretical contribution is an entropy-controlled discretization guarantee conditional on the learned-denoiser error. It is not an end-to-end sample-complexity theorem, and the final KL estimate may be loose when the learned predictor has large error.
>
> ### 4. Theoretical and practical status of LAS
>
> LAS is primarily a practical scheduling algorithm motivated by our KL decomposition and the exact loss identity in Proposition 5.1. The decomposition shows that optimizing $E_{\mathrm{disc}}$ alone ignores the learned-model error in $E_{\mathrm{apx}}$, while Proposition 5.1 expresses their schedule-dependent sum through the model’s prediction-loss profile.
>
> LAS reuses this training-loss information to compute a model-adaptive schedule at low post-training cost. We view it as a lightweight alternative to Align Your Steps, which requires a separate post-training procedure to estimate and optimize the scheduling objective using Monte Carlo estimation, including importance sampling. In contrast, LAS uses a loss profile already available from training, or cheaply estimated afterward, and solves the resulting finite-grid problem by dynamic programming.
>
> The regularized SNR axis and smoothness penalty for second-order solvers are practical choices for controlling high-SNR looseness and numerical instability, not consequences of Theorem 4.7. We now describe LAS as competitive overall, with its strongest improvements in low-step regimes, rather than uniformly superior on every metric.
>
> ### 5. Assumptions and constants
>
> We made the finite-second-moment, countable-support, and finite-$H_{1/2}$ assumptions explicit wherever the main theorem is summarized. We also state the additional sub-exponential information-content assumption required for the Shannon-entropy corollary.
>
> We removed the ambiguous formulation involving an unspecified lower threshold. For every finite $H_{1/2}$, the theorem now states
>
> $$|\operatorname{mmse}'(\gamma)|\leq\frac{\Psi(H_{1/2})}{\gamma^2},$$
>
> where $\Psi(h)=96h^2+64(h+1)$. Equivalently,
>
> $$|\operatorname{mmse}'(\gamma)|\leq\frac{96H_{1/2}^2+64(H_{1/2}+1)}{\gamma^2}.$$
>
> A separate simplified corollary states that, when $H_{1/2}\geq2$,
>
> $$|\operatorname{mmse}'(\gamma)|\leq\frac{144H_{1/2}^2}{\gamma^2}.$$
>
> Thus, the lower-order terms and threshold are explicit. We also state that the constants in the Shannon-entropy corollary depend on the sub-exponential information-content parameters.

---

### Review · Reviewer_4QHF · 2026-07-18

**Summary Of Contributions:**

This paper develops an information-theoretic analysis of reverse-diffusion discretization and represents the discretization error through the MMSE curve of the associated Gaussian channel. Under finite-moment and Rényi-entropy assumptions, it derives a dimension-free error bound and discusses the resulting choice of sampling schedule.

**Audience:**

Yes

**Audience Explanation:**

Dimension-free convergence analysis and sampling-schedule design for diffusion models are important and timely topics that are likely to be of interest.

**Claims And Evidence:**

No

**Claims Explanation:**

Several central technical points, particularly Proposition 3.1 and its proof in Appendix A, require further clarification or revision before the main convergence claims can be considered fully supported. I would be willing to revise this assessment if the authors satisfactorily address these issues.

**Requested Changes:**

Proposition 3.1 & Appendix A: The approximate process is initialized from $q$, whereas the exact reverse process is initialized from $p_T$. Therefore, the stated pathwise KL estimate appears to omit the initial divergence $\mathrm{KL}(p_T\Vert q)$. In Appendix A, the auxiliary processes used in the proof are initialized from the law of $X_T$, rather than from $q$. Thus, the path law considered in Appendix A does not seem to coincide with the approximate path law defined in the main text.

Appendix A: The localization argument uses $\tau_n\uparrow T_\delta$ under $P$ to justify convergence under a separate coupling of $\widetilde P_n$ and $\widetilde P$. This step appears to require further justification. Could the authors clarify the coupling and limiting argument?

Proposition C.4: The proof yields a bound containing both $H_{1/2}^2$ and $H_{1/2}+1$. Therefore, the stated estimate with a universal constant,

$$\mathbb{E}|Z'-Z|^4\leq C_4 t^2 H_{1/2}^2,$$

does not appear to follow as written.

Section 3 and equation (16): Since the reverse process is stopped at time $T_\delta=T-\delta$, its terminal distribution is the law of $Y_{T_\delta}=X_\delta$, namely the Gaussian-smoothed distribution $p_\delta=p*\mathcal N(0,\delta I_d)$, rather than the original target distribution $p$. Thus, equation (16) appears to provide a KL bound with respect to $p_\delta$. Could the authors clarify this distinction in the statement and interpretation of the main convergence result?

Additional minor comments:

Proposition 4.5: The statement that $\mathcal F_\gamma$ is equivalently generated by ${W_u\geq 1/\gamma}$ does not appear correct, since $X_u=Z+W_u$. The martingale property should instead be justified using the Markov property and the independence of future Brownian increments.

Proposition C.1: The MMSE derivative identity appears valid, but the proof would benefit from additional justification for the regularity of $u(s,y)$, the application of Itô's formula, and the interchange of differentiation and expectation.

---

> ### Author Response · Authors · 2026-07-20
>
> We thank the reviewer for the careful reading of the manuscript and for
> identifying these technical points. We have revised the relevant statements
> and proofs to clarify the initialization mismatch, the localization and
> coupling argument, the precise terminal target, and the regularity used in
> the MMSE analysis. Our detailed responses are provided below.
>
> ## 1. Proposition 3.1 and Appendix A: Initialization Mismatch
>
> We thank the reviewer for pointing this out. This issue has been addressed
> in the revision using the chain rule property of relative entropy.
>
> We now distinguish three path laws:
>
> - $\mathbb{P}$: the exact reverse process initialized from $p\_T$;
>
> - $\overline{\mathbb{P}}$: the approximate reverse dynamics initialized from $p\_T$;
>
> - $\widetilde{\mathbb{P}}$: the implemented approximate reverse dynamics initialized from $q$.
>
> The localization and Girsanov argument in Appendix A first compares
> $\mathbb{P}$ and $\overline{\mathbb{P}}$, which have the same initial distribution and
> differ only in their drifts. The laws $\overline{\mathbb{P}}$ and
> $\widetilde{\mathbb{P}}$ have the same approximate dynamics and differ only in
> their initial distributions.
>
> Since the conditional path laws given the initial state are the same for
> $\overline{\mathbb{P}}$ and $\widetilde{\mathbb{P}}$, the chain rule gives
> $$
> \operatorname{KL}\left(
>     \mathbb{P}
>     \\Vert
>     \widetilde{\mathbb{P}}
> \right)
> =
> \operatorname{KL}\left(
>     \mathbb{P}
>     \\Vert
>     \overline{\mathbb{P}}
> \right)
> +
> \operatorname{KL}\left(
>     p\_T
>     \\Vert
>     q
> \right).
> $$
>
> Combining this identity with the Girsanov estimate yields
> $$
> \operatorname{KL}\left(
>     \mathbb{P}
>     \\Vert
>     \widetilde{\mathbb{P}}
> \right)
> \leq
> \operatorname{KL}\left(
>     p\_T
>     \\Vert
>     q
> \right)
> +
> \frac{1}{2}
> \mathbb{E}\_{\mathbb{P}}
> \left[
>     \int\_0^{T\_\delta}
>     \lVert \delta\_s\rVert^2\mathrm{d}s
> \right].
> $$
>
> This is now the statement of Proposition 3.1 in the revision.
>
> ## 2. Appendix A: Localization and Coupling
>
> We thank the reviewer for pointing out an ambiguity in the localization
> argument. In the previous version, the stopping time $\tau\_n$ was first
> defined using the exact process under $\mathbb P$, while the statement
> $\tau\_n \uparrow T\_\delta$ was subsequently used on a different probability
> space carrying a coupling of $\widetilde {\mathbb P}\_n$ and $\overline{\mathbb P}$. The
> ${\mathbb P}$-almost-sure convergence alone does not justify this latter step.
>
> In the revised proof, we define $\tau\_n$ as a nonanticipative functional on
> the canonical path space. Thus, on any probability space carrying a random
> path $\xi$, the corresponding random time is $\tau\_n(\xi)$.
>
> $$
> \\tau\_n(\\xi):= \\inf\\left\\{
> t \\in [0,T\_\\delta] :
> \\int\_0^t
> \\left\\lVert
> \\frac{m\_{T-s}(\\xi(s))-\\widehat{m}\_{T-s\_{k-1}}(\\xi(s\_{k-1}))}{T-s}
> \\right\\rVert_2^2
> ds
> \\geq n
> \\right\\}
> \\wedge T\_\\delta.
> $$
>
> In the coupling,
> we set
> $$
>     \sigma\_n := \tau\_n(\widetilde\xi),
> $$
> where $\widetilde\xi$ has law $\overline{\mathbb P}$, and construct
> $\widetilde\xi^n$ so that it agrees with $\widetilde\xi$ up to $\sigma\_n$
> and follows the exact drift thereafter. Nonanticipativity then gives
> $$
>     \tau\_n(\widetilde\xi^n)
>     =
>     \tau\_n(\widetilde\xi)
>     =
>     \sigma\_n.
> $$
> Consequently, $\widetilde\xi^n$ solves the localized SDE obtained from
> Girsanov's theorem, and uniqueness in law identifies
> $\mathcal L(\widetilde\xi^n)=\widetilde {\mathbb P}\_n$.
>
> We also show directly that the drift-mismatch energy is finite along every
> continuous path. Indeed, $T-t\geq\delta$, the Gaussian-smoothed denoiser
> $m\_t(x)$ is jointly locally bounded for $t\in[\delta,T]$, and the learned
> denoiser is evaluated only at finitely many grid points. Therefore
> $$
>     \int\_0^{T\_\delta}
>     \left\|
>         \beta\_t(\omega\_t)-\widetilde\beta\_t(\omega\_t)
>     \right\|^2dt
>     <\infty
> $$
> for every continuous path $\omega$. It follows that
> $\sigma\_n=T\_\delta$ for all sufficiently large $n$, almost surely under the
> coupling measure. Hence $\widetilde\xi^n=\widetilde\xi$ eventually as entire
> paths, so that $\widetilde {\mathbb P}\_n\Rightarrow\widetilde {\mathbb P}$. Lower
> semicontinuity of relative entropy then yields the claimed bound.
>
> This argument no longer transfers a $P$-almost-sure statement to the
> coupling space, and the auxiliary terminal-time truncation used in the
> previous proof is unnecessary.

---

> > ### Author Response · Authors · 2026-07-20
> >
> > ## 3. Proposition C.4
> >
> > We agree with the reviewer. The direct calculation gives
> > $$
> > \mathbb{E}\left[
> >     \lVert Z'-Z\rVert^4
> > \right]
> > \leq
> > 96t^2H\_{1/2}^2
> > +
> > 64t^2\bigl(H\_{1/2}+1\bigr),
> > $$
> > rather than a universal multiple of $t^2H\_{1/2}^2$ without an additional
> > assumption.
> >
> > We have therefore revised Proposition C.4 to state the exact estimate and
> > introduced
> > $$
> > \Psi(h)
> > :=
> > 96h^2+64(h+1).
> > $$
> >
> > Accordingly, Theorem 4.7 now states
> > $$
> > \left|
> >     \operatorname{mmse}'(\gamma)
> > \right|
> > \leq
> > \frac{\Psi(H\_{1/2})}{\gamma^2}.
> > $$
> >
> > The simplified estimate
> > $$
> > \left|
> >     \operatorname{mmse}'(\gamma)
> > \right|
> > \leq
> > \frac{144H\_{1/2}^2}{\gamma^2}
> > $$
> > is now stated separately under the additional condition $H\_{1/2}\geq 2$.
> > Indeed, under this condition,
> > $$
> > 96H\_{1/2}^2
> > +
> > 64\bigl(H\_{1/2}+1\bigr)
> > \leq
> > 144H\_{1/2}^2.
> > $$
> >
> > The proposition and the subsequent results have been updated accordingly.
> >
> > ## 4. Section 3 and the Terminal Target Distribution
> >
> > We agree with the reviewer. The result controls the terminal sampling error relative to the Gaussian-smoothed distribution $p\_\delta$, which is the exact target at the stopping time $T\_\delta$. It does not directly control the KL divergence relative to the original distribution $p$. The discrepancy between $p\_\delta$ and $p$ is a separate endpoint-smoothing issue and is not quantified by the present theorem.
> >
> > ## 5. Proposition 4.5
> >
> > We thank the reviewer for pointing this out. We agree that the statement that
> > the filtration is equivalently generated by the Brownian variables is
> > incorrect.
> >
> > We have removed this statement and redefined the filtration as
> > $$
> > \mathcal{F}\_\gamma
> > :=
> > \sigma\left(
> >     X\_{1/\gamma'}:
> >     \frac{1}{T}\leq \gamma'\leq \gamma
> > \right),
> > \qquad
> > \gamma\in
> > \left[
> >     \frac{1}{T},
> >     \frac{1}{\delta}
> > \right].
> > $$
> > The remainder of the proof of Proposition 4.5 are unaffected.
> >
> > ## 6. Proposition C.1
> >
> > We thank the reviewer for this comment. Since
> >
> > $$
> > X\_t=Z+\\sqrt{t}\\varepsilon,
> > \\qquad
> > \\varepsilon\\sim\\mathcal{N}(0,I\_d),
> > $$
> >
> > the density $p\_t$ of $X\_t$ is the convolution of $p$ with a nondegenerate
> > Gaussian density. Therefore, for every $t>0$, $p\_t$ is strictly positive and
> > belongs to $C^\infty$. Consequently, $\log p\_t$ also belongs to
> > $C^\infty$.
> >
> > By Tweedie's formula,
> >
> > $$
> > m\_t(x)=\mathbb{E}\left[
> >     Z
> >     \middle|
> >     X\_t=x
> > \right]
> > =
> > x+t\nabla\log p\_t(x).
> > $$
> >
> > Moreover,
> >
> > $$
> > u(s,y)=\mathbb{E}\left[
> >     Y\_T
> >     \middle|
> >     Y\_s=y
> > \right]
> > =
> > m\_{T-s}(y).
> > $$
> >
> > Since the reverse process is considered only for
> > $$
> > s\leq T-\delta,
> > $$
> > we have
> > $$
> > T-s\geq \delta>0.
> > $$
> >
> > It follows that $u$ is smooth throughout the time interval under
> > consideration, which justifies the application of It\^{o}'s formula. We have
> > added this clarification to the proof of Proposition C.1.
> >
> > We thank the reviewer again for identifying these issues. The revisions
> > clarify the decomposition of the pathwise KL error, the common-space
> > localization argument, the precise Gaussian-smoothed terminal target, and
> > the regularity used in the MMSE calculation. These changes do not alter the
> > MMSE representation of the discretization error or the resulting
> > dimension-free entropy-dependent bound.